# MGD³: Mode-Guided Dataset Distillation using Diffusion Models

## Abstract

Dataset distillation aims to distill a smaller training dataset from a larger one so that a model trained on this smaller set performs similarly to one trained on the full dataset. Traditional methods are costly and lack sample diversity. Recent approaches utilizing generative models, particularly diffusion models, show promise in capturing data distribution, but they often oversample prominent modes, limiting sample diversity. To address these limitations in this work, we propose a mode-guided diffusion model. Unlike existing works that fine-tune the diffusion models for dataset distillation, we propose to use a pre-trained model without the need for fine-tuning. Our novel approach consists of three stages: *Mode Discovery*, *Mode Guidance*, and *Stop Guidance*. In the first stage, we discover distinct modes in the data distribution of a class to build a representative set. In the second stage, we use a pre-trained diffusion model and guide the diffusion process toward the discovered modes to generate distinct samples, ensuring intra-class diversity. However, mode-guided sampling can introduce artifacts in the synthetic sample, which affect the performance. To control the fidelity of the synthetic dataset, we introduce the stop guidance. We evaluate our method on multiple benchmark datasets, including ImageNette, ImageIDC, ImageNet-100, and ImageNet-1K; Our method improved $4.4\%$, $2.9\%$, $1.6\%$, and $1.6\%$ over the current state-of-the-art on the respective datasets. In addition, our method does not require retraining of the diffusion model, which leads to reduced computational requirements. We also demonstrate that our approach is effective with general-purpose diffusion models such as Text-to Image Stable Diffusion, showing promising performance towards eliminating the need for a pre-trained model in the target dataset. Our source code is available in this anonymized repository.

## 1 Introduction

The rapid advancements in machine learning are marked by a trend towards increasingly large datasets and models to achieve state-of-the-art performance. However, this trend presents significant challenges for researchers constrained by limited computation and storage resources. In response, the research community started to focus on developing techniques to address these limitations. While model pruning (Liu et al., 2017; He et al., 2019; Ding et al., 2019; Sharma & Foroosh, 2022) and quantization (Xu et al., 2023; Chauhan et al., 2023; Chen et al., 2021; Wu et al., 2016) are introduced to improve the model efficiency, core set selection and dataset distillation (Wang et al., 2018; Liu et al., 2022) have emerged as the prominent techniques to reduce the size of the datasets. Core set selection (Chen et al., 2010; Castro et al., 2018; Welling, 2009; Rebuffi et al., 2017) based approaches were initially introduced for building condensed datasets, which involves selecting a few prototypical examples from the original dataset to build the smaller dataset. However, these approaches are limited to choosing the samples from the original dataset, which considerably restricts the expressiveness of the condensed dataset. Dataset distillation removes this restriction.

The task of data set distillation is to distill information from a large training dataset into a smaller dataset with few synthetic samples such that a model trained on the smaller data set achieves performance comparable to the model trained on the complete data set. Most of the existing data set distillation methods (Cazenavette et al., 2022b;a; Zhao & Bilen, 2023) follow the data matching framework, where the distilled dataset is updated to mimic the influence of the original dataset on

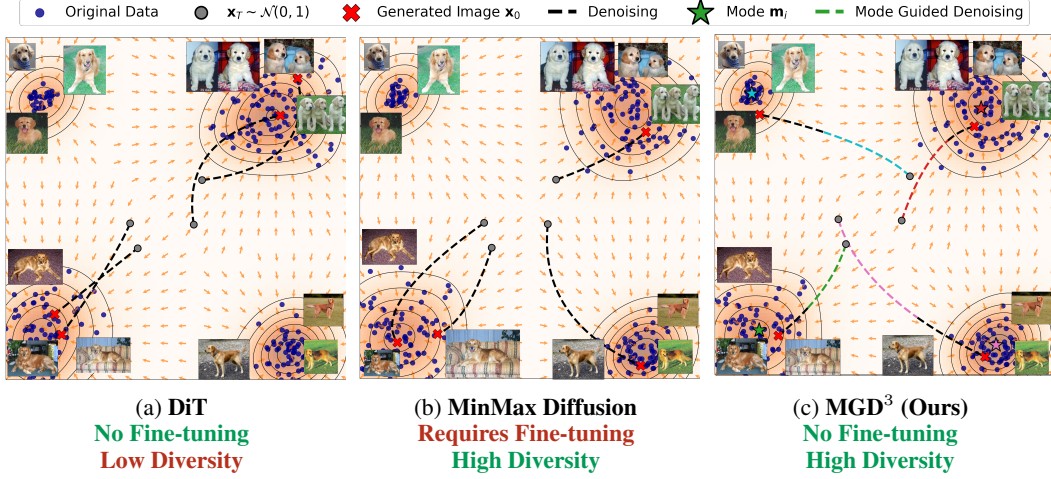

Figure 1: Overview of the gradient field (score function) during the denoising process in latent diffusion for a specific class $c$. The original data distribution, marked by blue dots, shows denser regions (orange shadow) in the gradient field. To generate an image $\hat{X}_i$, noise $x_T{}^i \sim N(0, \mathbf{I})$ is sampled. In (a), a pre-trained diffusion model demonstrates imbalanced mode likelihood, leading to limited sample diversity and repeated modes. (b) shows MinMax Diffusion, which fine-tunes the model to enhance diversity by balancing mode likelihoods, but still faces redundancies based on initial noise conditions. (c), the proposed method introduces mode guidance in the denoising process (green and red traces), directing samples towards distinct modes (stars). After $k$ steps of guidance, it transitions to unguided denoising (black trace), achieving high diversity and consistency without the need for fine-tuning.

model training. These methods aim to minimize the distribution gap between the original and distilled datasets by considering different aspects, such as model parameters, long-range training trajectories, or feature distribution. However, these methods are far from optimal, as they need to repeat the execution of their method to synthesize distilled datasets of different sizes. In addition, they tend to generate out-of-distribution samples (Su et al., 2024). To address these challenges, recent works (Wang et al., 2023; Su et al., 2024; Zhang et al., 2023) propose storing the knowledge of the dataset into the parameters of a generative model instead of directly condensing it into a smaller synthetic set. Once trained, the same generative model can generate synthetic datasets of varied sizes.

Among the generative models, diffusion models (Ho et al., 2020) are known for their impressive capabilities in image synthesis. These models achieve perceptual quality comparable to GANs while offering higher distribution coverage, as evidenced by (Dhariwal & Nichol, 2021b). However, they tend to concentrate on denser regions (modes) of the data distribution, resulting in a synthetic dataset that, while representative, often lacks the full diversity of the original data (Gu et al., 2024) (refer to Figure 1a). Previous works (Gu et al., 2024) address this by explicitly retraining the diffusion model to generate representative and diverse samples. With this training, the samples are more likely to be generated from different modes of a class (See Figure 1b). In contrast, we propose a novel approach that extracts diverse samples from a pre-trained diffusion model already trained on the target dataset, without the need for further model retraining.

Our approach aims to distill a dataset with representative and diverse samples without re-training the diffusion model. Our method first estimates the prevalent modes in the data with the *Mode Discovery* stage. Then, we ensure diversity by guiding each sample to a different mode with our *Mode Guidance*. However, guiding the sample to an estimated mode may compromise the quality of the synthetic sample. To address this, we introduce *Stop Guidance* to maintain the quality of the synthetic data (see Figure 1c).

In summary, the key contributions and results of our work are as follows:

- We propose a novel approach for dataset distillation that leverages a pre-trained diffusion model to extract a representative dataset without requiring model retraining or fine-tuning.

- We propose *Mode Guidance with Stop Guidance*, a novel sampling technique that systematically samples from different modes of the data distribution while preserving sample quality. Our approach enables controlled diversity sampling for dataset synthesis by guiding the denoising process across distinct class regions.

- Our method demonstrates improved or comparable performance to the current state-of-the-art while eliminating the need for fine-tuning the diffusion model. For instance, our approach achieves competitive results by even using a text-to-image model like Stable Diffusion, highlighting its adaptability and effectiveness across various diffusion models without additional fine-tuning.

## 2 RELATED WORK

Dataset distillation has received increased interest in recent years due to its applications in continual learning (Zhao et al., 2021; Zhao & Bilen, 2021; 2023), privacy-preserving datasets (Sucholutsky & Schonlau, 2021; Li et al., 2020), neural architecture search (Zhao et al., 2021; Zhao & Bilen, 2021), and model explainability (Loo et al., 2022). Prior works have explored the problem of dataset distillation and show how challenging it is to encapsulate datasets in a limited set of examples. Initially, this task was approached using non-generative models, then using generative priors, and, more recently, with generative models. Below, we discuss works belonging to these categories in detail.

### 2.1 NON-GENERATIVE DATASET DISTILLATION METHODS

Dataset distillation condenses information from a large dataset into a smaller one with synthetic images, to enable model training on the smaller dataset with performance comparable to that of the full dataset. Various methods have emerged to minimize the performance gap between original and distilled datasets through techniques such as dataset matching, parameter matching, and gradient matching.

Initially, parameter matching (Zhao et al., 2021) was proposed, aligning the neural network weights trained on synthetic data with those from the original dataset. However, this bi-level optimization approach was time-consuming and unscalable. Gradient matching was later favored due to the high dimensionality of parameter space, leading to inefficiencies. Further advances included feature matching, which improves efficiency by eliminating dependence on bi-level optimization (Zhao & Bilen, 2023). Additionally, long-range matching techniques were introduced by (Cazenavette et al., 2022b) through matching training trajectories (MTT), optimizing network parameters after multiple training iterations to better synthesize relevant features for the training updates.

Recent advancements have introduced generative priors into the optimization process. GAN-IT (Zhao & Bilen, 2022) shifted the focus from the pixel space to latent codes of pre-trained GANs, optimizing these codes rather than working directly in image space. GLaD (Cazenavette et al., 2023) built on this by incorporating generative priors with StyleGAN for high-resolution datasets, yielding images that more closely match the dataset distribution and improve performance. H-GLaD (Zhong et al., 2024) further enhanced this by focusing on deeper feature layers for hierarchical optimization. Additionally, LD3M (Moser et al., 2024) utilized a latent diffusion model to optimize synthetic datasets directly in the model's latent space, improving performance by refining latent codes through the denoising and diffusion processes. Despite their success on small-resolution datasets, these methods struggle with high-resolution datasets (e.g., 256 x 256, 20 images per class), often being computationally expensive and less efficient, leading to the emergence of more effective distillation methods from generative models (Gu et al., 2024; Su et al., 2024; Zhang et al., 2023).

### 2.2 GENERATIVE DATASET DISTILLATION METHODS

While generative priors have been getting attention, recent work (Zhang et al., 2023; Gu et al., 2024; Su et al., 2024) has started focusing on distilling datasets using generative models. Unlike approaches that use generative priors only to optimize the latent vectors of a generative model, generative dataset distillation aims to explicitly train a generative model that can synthesize a distilled dataset. Recent work, (Zhang et al., 2023) introduced one of the initial approaches using generative models for

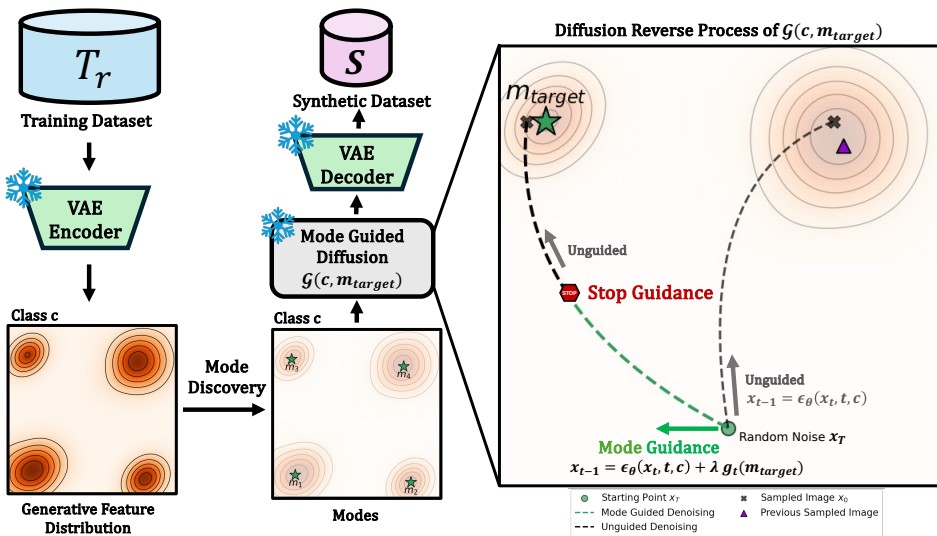

Figure 2: Overview of the proposed method for distilled dataset synthesis using a diffusion model. Our approach consists of three key stages: *Mode Discovery*, *Mode Guidance*, and *Stop Guidance*. In the Mode Discovery stage, we estimate the $N$ modes of the original dataset within the generative space of the latent diffusion model. Given a mode $m_k$ and a class $c$, the Mode-Guided Diffusion process directs the generation toward the specified mode $m_k$. This guidance is applied for $t_{SG}$ steps until the Stop Guidance stage, after which unguided diffusion takes over. During sampling, mode guidance ensures that images from the desired mode $m_k$ are generated using the pre-trained diffusion model. If no guidance is applied, the generation follows the unguided (grey) path, which can lead to redundancies in the dataset.

dataset distillation. They proposed to use a class-conditional GAN with a learnable codebook for each image in the class and optimize it with GAN loss to generate realistic-looking examples, intra-class diversity loss to introduce diversity, inter-class discriminative loss to promote representativeness, and matching loss to facilitate dataset condensation. Similarly, (Gu et al., 2024) brought these ideas to diffusion models by fine-tuning a pre-trained diffusion model with representative and diversity losses to improve the performance. Later work (Su et al., 2024) proposed the $D^4M$ model, which uses Text-to-Image Stable diffusion as a generator and proposes a disentangled diffusion that substitutes random noise for noisy modes as starting noise when sampling distilled samples. However, the high level of noise in the initial denoising steps often results in outputs that are neither diverse nor fully representative. Additionally, the model relies on a pre-trained classifier to assign soft labels during training with the distilled dataset. While previous generative models have primarily focused on developing new loss functions to improve representativeness and diversity, our work introduces a training-free approach that achieves both properties without the need for complex losses or additional training.

## 3  METHOD

Our approach for obtaining diverse and representative examples for each class involves leveraging a diffusion model trained on the target dataset. The main idea is to sample from the more likely regions in the data distribution, called modes, during the reverse process. These modes are regions where the samples have similar appearances, and they are representative of the class. However, diffusion models tend to sample from the more likely modes, which creates an issue when there are fewer dominant modes for a particular class than the number of images per class (IPC) in the dataset. Previous work (Gu et al., 2024) requires fine-tuning the diffusion model with additional representative and diversity losses to balance the likelihood of all modes; this approach requires more training and still has a high chance of redundancy while sampling.

Our three-stage approach, shown in Figure 2, eliminates the need for fine-tuning while maintaining mode diversity. In the first stage, known as mode discovery, we estimate a diverse set of modes

for each class in the dataset. After estimating the modes, the second stage involves controlling the reverse process using our proposed mode guidance to sample from the estimated mode distribution. During sampling, the guidance is applied until the stop guidance is triggered, which helps to control the quality of the synthetic sample.

In the mode discovery stage, the main objective is to identify the $N$ modes of a specific class in the original dataset distribution. This discovery is performed using the original dataset in the latent space of the VAE encoder ($V_{enc}$). The motivation for this approach is that the generative space captures the overall content of the image rather than discriminative features, which can be limited to specific textures in the image. Any clustering algorithm can be used to estimate the modes for a particular class; in our approach, we use K-Means. Once the modes are identified, our goal is to sample images from these estimated modes. In the next section, we explain how to guide the denoising process of diffusion and guarantee generation from the specified modes. The rest of this section is organized as follows. Next, we cover some preliminaries, followed by Mode Guidance and Stop Guidance.

## 3.1 PRELIMINARIES

**Dataset Distillation:** Given a large-scale dataset with the training set $\mathcal{T}_r = \{(X_i, y_i)\}_{i=1}^{N_{\mathcal{T}}}$, the goal of dataset distillation is to build a smaller synthetic dataset $\mathcal{S} = \{(\tilde{X}_i, \tilde{y}_i)\}_{i=1}^{N_{\mathcal{S}}}$, where $N_{\mathcal{S}} << N_{\mathcal{T}}$ and $X_i, \tilde{X}_i$ are the original and synthetic images with the corresponding class labels $y_i, \tilde{y}_i$. In addition, the model $\theta_{\mathcal{T}}$ trained on the original training set should achieve similar test performance as the model $\theta_{\mathcal{S}}$ trained on the smaller synthetic dataset; i.e. if $\mathcal{A}$ is the accuracy of a model on the test set ($\mathcal{T}_e$), then $\mathcal{A}(\theta_{\mathcal{T}}) \sim \mathcal{A}(\theta_{\mathcal{S}})$. During the evaluation, the size of the distilled dataset $N_{\mathcal{S}}$, is set based on the distillation budget, denoted by IPC, the number of images allocated per class.

Our approach builds on the foundations of prior generative models, such as (Gu et al., 2024; Su et al., 2024; Zhang et al., 2023), which address dataset distillation by approximating the dataset distribution through sampling diverse and representative instances. This line of work can be characterized as dataset distillation through dataset matching. Where the objective of the distillation is defined as

$$\left\| \mathbb{E}_{x \sim P(\mathcal{D})}\big[l(\phi_{\theta^{\mathcal{T}}}(x), y)\big] - \mathbb{E}_{x \sim P(\mathcal{D})}\big[l(\phi_{\theta^{\mathcal{S}}}(x), y)\big] \right\| < \epsilon$$

Where, $P(\mathcal{D})$ is the real data distribution, $\phi_{\theta^{\mathcal{T}}}$ and $\phi_{\theta^{\mathcal{S}}}$ are models trained on the original dataset $\mathcal{T}$ and the distilled dataset $\mathcal{S}$ respectively. Note, that this formulation is similar to the coreset methods, however, the used of generative models remove the restriction of choosing a sample from the original dataset, meaning that can approximate the original dataset better.

**Diffusion Model:** The denoising probabilistic diffusion model (DDPM) is a generative model, $\mathcal{G}$, that learns a mapping between Gaussian noise and the data distribution through a series of T denoising steps. $\mathcal{G}$ assumes a Markov chain that gradually adds noise to a sample $x_0$ in the data distribution, which is called the forward process. The forward process of $\mathcal{G}$ is defined as $q(x_t|x_{t-1}) = N(\sqrt{1 - \beta_t}x_{t-1}, \beta_t\mathbf{I})$, where $\beta_t$ is the variance schedule for the time step t. In practice, this is done using the reparametrization trick $x_t = \sqrt{\bar{\alpha}}x_0 + \sqrt{1 - \bar{\alpha}}\epsilon_t$, where $\epsilon_t \sim N(0, \mathbf{I})$.

Image generation is done by the reverse process of $\mathcal{G}$, where $\epsilon_\theta$ is the noise prediction network, trained to reverse the Markov chain $p_\theta(x_{t-1}|x_t) = N(\mu_\theta(x_t), \Sigma_\theta(x_t))$, where $\theta$ corresponds to the parameters of the model and $\mu_\theta(x_t), \Sigma_\theta(x_t)$ are the $\mu$ and $\Sigma$ predictions of the denoising models. $\mu_\theta(x_t)$ is computed as follows:

$$\mu_\theta(x_t) = \frac{1}{\sqrt{1 - \beta_t}}\Big(x_t - \frac{1}{\sqrt{1 - \alpha}}\epsilon_\theta(x_t, t)\Big) + \sigma_t\mathbf{z} \tag{1}$$

where $\mathbf{z} \sim N(0, 1)$ and $\sigma_t$ is the variance schedule. $\epsilon_\theta(x_t, t)$ is the output of the noise prediction network that is trained to predict the added noise with the simple loss defined as

$$\mathcal{L}_\theta = ||\epsilon_\theta(x_t, t) - \epsilon_t||^2 \tag{2}$$

After training, $\mathcal{G}$ can generate samples by sampling from the noise distribution and running the reverse process. In this work, we use a class-conditioned diffusion model $\mathcal{G}_c$, where the output of the noise prediction network conditioned with the class $c$, is denoted as $\epsilon_\theta(x_t, t, c)$.

**Latent Diffusion:** The latent diffusion model is a variation of the DDPM that applies the forward and denoising process in the latent space representation of a VAE ($V_{enc}$). The model is trained to

denoise a noisy latent space representation at each time $t$. After denoising the latent representation, the VAE decoder ($V_{dec}$) generates the final image. Note that the latent space is smaller and more semantic than the image space.

**Diffusion Guidance:** The sampling process of DDPM is equivalent to score-based generative models by interpreting $\epsilon_\theta(x_t, t) = -\sqrt{\alpha}\nabla_x \log p(x_t)$, where $\nabla_x \log p(x_t)$ is an estimation of the score function. For the case of class-conditioned generation, by using Bayes' rule the score function can be derived as:

$$\nabla_x \log p(x_t|c) = \nabla_x \log p(x_t) + \nabla_x \log p(c|x_t) \tag{3}$$

where $\nabla_x \log p(c|x_t)$ is the gradient of the class-conditional log-likelihood. It's important to note that $\nabla_x \log p(c|x_t)$ represents the drift of the diffusion process towards the distribution of the class $c$. In (Dhariwal & Nichol, 2021a), a classifier is used to estimate the class-conditional log-likelihood and use it as a guidance signal to direct the diffusion process towards the desired class. Later, (Ho & Salimans, 2022) suggested using a combination of unconditional generation and conditional diffusion (eq. 4) to remove the dependency on the classifier and demonstrated improved results and called this classifier-free guidance. Classifier-free guidance is defined as

$$\tilde{\epsilon}_\theta(x_t, t, c) = (1 - w) \cdot \epsilon_\theta(x_t, t, c) - w \cdot \epsilon_\theta(x_t, t) \tag{4}$$

where the $w$ is the guidance scale that controls how strong the guidance is applied.

### 3.2 Mode Guidance

In this section, we describe the method used for image synthesis with mode guidance. Our goal is to generate high-quality images belonging to a specific class mode. Given a class $c$ and a set of discovered modes for that class denoted as $\mathbf{M_c} = \{m_1, ..., m_{IPC}\}$, our method computes the mode guidance score for a particular mode $m_i$ using the following equation:

$$\mathbf{g}_t = (m_i - \hat{x_0}^t), \tag{5}$$

where $\hat{x_0}^t$ is the predicted denoised latent feature at timestep $t$ during the reverse process. We apply this guidance signal at the $x_t$ timestep as follows:

$$\hat{\epsilon}_\theta(x_t, t, c) = \tilde{\epsilon}_\theta(x_t, t, c) + \lambda \cdot \mathbf{g}_t \cdot \sigma_t, \tag{6}$$

where $\lambda$ is a scalar that controls the strength of the guidance signal.

To synthesize an image from a particular mode $m_i$, the diffusion model $\mathcal{G}$ calculates the mode guidance score at each iteration of the reverse process using Eq.6. This score represents the direction from the predicted value to the mode $m_i$. The guidance signal is then added to the noise function at the appropriate time step in the diffusion process. By adjusting the strength of the guidance signal, we can regulate the impact of the mode on the generated image.

### 3.3 Stop Guidance

The reverse process of diffusion can be divided into three stages: the chaotic stage (first $20\%$), the semantic stage ($20\%$ to $50\%$), and the refinement stage (last $50\%$) (Yu et al., 2023). We hypothesize that mode guidance is unnecessary during the refinement stage because we believe that the purpose of mode guidance is to guide the synthetic image towards the mode in the high semantic space. Also, during our initial experiments, we observed that providing strong guidance to a particular mode $m_i$ often results in a loss of class fidelity and the presence of image artifacts (See Figure 4b $t_{SG} = 0$). Therefore, we introduce the stop guidance to mitigate losing class fidelity and image artifacts. The stop guidance consists of giving a stop timestep $t_{SG}$, the stop guidance involves setting $\lambda$ to zero in Eq. 6, when $t < t_{SG}$ in the reverse process. In the experiment section, we provide an ablation for the timestep ($t_{SG}$) when guidance ceases to exist.

## 4 EXPERIMENTS

**Datasets and evaluation.** To assess our approach's effectiveness, we thoroughly examined the available benchmarks for distilling high-resolution datasets ($256 \times 256$). The datasets we evaluate include ImageNet-1K, ImageNet-100, ImageNetIDC, ImageNette, and ImageNet-A to ImageNet-E. Additionally, we included results from ImageWoof in the supplemental material. We used two protocols for evaluation: a hard-label protocol and a soft-label protocol.

The hard-label protocol generates a dataset with its corresponding class labels, trains a network from scratch, and evaluates the network on the original test set. This process is repeated three times for target architectures, and the accuracy mean and standard deviation are reported. Random resize-crop and CutMix are applied as augmentation techniques during the target network's training. For more detailed technical information about the protocol, please refer to (Gu et al., 2024). Similar to the existing literature, we evaluate our model in various IPCs ranging from 10 to 100. This protocol was used to evaluate ImageNet-100, ImageNette, and ImageNetIDC datasets.

In soft-label protocol, region-based soft-labels are generated with a pre-trained network as proposed by (Sun et al., 2024). The region-based soft-labels $y_{i,m}$ are generated as follows: $y_{i,m} = \phi_{\mathcal{T}}(x_{i,m})$, where $\phi_{\mathcal{T}}$ is the pretrained model and $x_{i,m}$ is the $m$-th crop of the $i$-th image. When training a model $\phi_{\mathcal{S}}$ on the distilled dataset the objective loss is $\mathcal{L} = -\sum_j \sum_m y_{j,m} \log \phi_{\mathcal{S}}(x_{j,m})$. For ImageNet-1k evaluation, we follow this protocol. Similarly to (Sun et al., 2024; Gu et al., 2024), we used ResNet-18 as a teacher and student network architecture for this setup.

**Baselines.** We are comparing two baselines: 1) The pre-trained DiT XL/2, which represents diffusion models without mode guidance. 2) MinMax diffusion with DiT XL/2, which represents a scenario where the diffusion model is fine-tuned to encourage diversity and representativeness. For the ImageNette and IDC datasets, we used a class-conditioned Latent Diffusion Model (LDM) (Rombach et al., 2022) trained on ImageNet-1k to compare the U-Net architecture versus Transformer architecture in the diffusion model. In our experiments, both DiT and LDM use the DDPM sampler. Additional results with the DDIM sampler are provided in the supplemental material. To compare with $D^4M$ (Su et al., 2024) in our hard label protocol, we applied a disentangled diffusion stage from $D^4M$ without including the soft labels by the Training Time Matching.

**General-purpose Diffusion Model.** Our method is adaptable to various diffusion models, with optimal performance observed when the model is pre-trained on the target dataset. To assess the generalizability of our approach, we tested it on a general-purpose diffusion model, specifically a text-to-image diffusion model. This evaluation poses challenges due to the potential mismatch between the model's training data and the target dataset. For this setup, the baseline was the text-to-image Stable Diffusion model without mode guidance, allowing us to demonstrate the impact of integrating mode guidance on the generated dataset. For sampling, we use the class names as a text prompt.

**Implementation details.** Our pre-trained model $\mathcal{G}$ is DiT-XL/2 trained on ImageNet, and the image size is 256 x 256. We use the sampling strategy described in (Peebles & Xie, 2023), which uses 50 sampling steps using classifier-free guidance with a guidance scale of $4.0$. For Mode Guidance, we set $\lambda$ to $0.1$, and in our experiments, we used stop guidance $t_{SG} = 25$. We use $k-$means to perform mode discovery; we set $k = IPC$. We used a single NVIDIA RTX A5000 GPU with 24GB VRAM to run our experiments.

### 4.1 COMPARISON WITH STATE-OF-THE-ART METHODS

In this study, we compare our method with current state-of-the-art (SOTA) techniques on various image datasets and architectures. Our method significantly outperforms previous approaches across various benchmark datasets and target architectures.

**ImageNette and ImageIDC.** In the ImageNette dataset, our method with DiT achieved performance gains of $4.4\%$, $4.4\%$, and $2.9\%$ in IPC 10, 20, and 50, respectively, outperforming the previous SOTA methods (See Table 1). Similarly, in the ImageIDC dataset, our method demonstrated improvements of $2.8\%$, $2.9\%$, and $2.5\%$ in IPC 10, 20, and 50, respectively, compared to the previous SOTA. Table 1 shows that our method consistently improves on three diffusion models: DiT, LDM, and Stable Diffusion. Notably, in the general purpose (Text-to-Image) evaluation, Stable Diffusion

| IPC | Nette | | | IDC | | |
|---|---|---|---|---|---|---|
| | 10 | 20 | 50 | 10 | 20 | 50 |
| Random | $54.2_{\pm 1.6}$ | $63.5_{\pm 0.5}$ | $76.1_{\pm 1.1}$ | $48.1_{\pm 0.8}$ | $52.5_{\pm 0.9}$ | $68.1_{\pm 0.7}$ |
| DM (Zhao & Bilen, 2023) | $60.8_{\pm 0.6}$ | $66.5_{\pm 1.1}$ | $76.2_{\pm 0.4}$ | $52.8_{\pm 0.5}$ | $58.5_{\pm 0.4}$ | $69.1_{\pm 0.8}$ |
| MinMaxDiff (Gu et al., 2024) | $62.0_{\pm 0.8}$ | $66.8_{\pm 0.4}$ | $76.6_{\pm 0.2}$ | $53.1_{\pm 0.2}$ | $59.0_{\pm 0.4}$ | $69.6_{\pm 0.2}$ |
| LDM (Rombach et al., 2022) | $60.3_{\pm 3.6}$ | $62.0_{\pm 2.6}$ | $71.0_{\pm 1.4}$ | $50.8_{\pm 1.2}$ | $55.1_{\pm 2.0}$ | $63.8_{\pm 0.4}$ |
| LDM+ Disentangled Diffusion ($D^4M$ (Su et al., 2024)) | $59.1_{\pm 0.7}$ | $64.3_{\pm 0.5}$ | $70.2_{\pm 1.0}$ | $52.3_{\pm 2.3}$ | $55.5_{\pm 1.2}$ | $62.7_{\pm 0.8}$ |
| LDM+ **MGD$^3$ (Ours)** | $61.9_{\pm 4.1}$ | $65.3_{\pm 1.3}$ | $74.2_{\pm 0.9}$ | $53.2_{\pm 0.2}$ | $58.3_{\pm 1.7}$ | $67.2_{\pm 1.3}$ |
| DiT (Peebles & Xie, 2023) | $59.1_{\pm 0.7}$ | $64.8_{\pm 1.2}$ | $73.3_{\pm 0.9}$ | $54.1_{\pm 0.4}$ | $58.9_{\pm 0.2}$ | $64.3_{\pm 0.6}$ |
| DiT+ Disentangled Diffusion ($D^4M$ (Su et al., 2024)) | $60.4_{\pm 3.4}$ | $65.5_{\pm 1.2}$ | $73.8_{\pm 1.7}$ | $51.1_{\pm 2.4}$ | $58.0_{\pm 1.4}$ | $64.1_{\pm 2.5}$ |
| DiT + **MGD$^3$ (Ours)** | $\mathbf{66.4}_{\pm 2.4}$ | $\mathbf{71.2}_{\pm 0.5}$ | $\mathbf{79.5}_{\pm 1.3}$ | $\mathbf{55.9}_{\pm 2.1}$ | $\mathbf{61.9}_{\pm 0.9}$ | $\mathbf{72.1}_{\pm 0.8}$ |
| | General Purpose Diffusion | | | | | |
| Stable Diffusion (Text-to-Image) (Rombach et al., 2022) | $46.4_{\pm 2.8}$ | $54.6_{\pm 2.0}$ | $60.6_{\pm 2.4}$ | $40.5_{\pm 1.1}$ | $43.9_{\pm 2.1}$ | $52.1_{\pm 1.2}$ |
| Stable Diffusion (Text-to-Image) + **MGD$^3$ (Ours)** | $\mathbf{57.3}_{\pm 2.2}$ | $\mathbf{63.3}_{\pm 2.6}$ | $\mathbf{74.4}_{\pm 1.4}$ | $\mathbf{48.5}_{\pm 0.8}$ | $\mathbf{51.6}_{\pm 1.3}$ | $\mathbf{60.2}_{\pm 3.1}$ |

Table 1: Performance comparison with pre-trained diffusion models and state-of-the-art methods on ImageNet subsets. The results are obtained on ResNet-10 with average pooling. The best results are marked as **bold**. Accuracy is the evaluation metric presented here.

underperforms all the methods; however, when combined with our method, the performance gap is significantly reduced.

| | 10 (0.8%) | | | 20 (1.6%) | | |
|---|---|---|---|---|---|---|
| | ConvNet-6 | ResNetAP-10 | ResNet-18 | ConvNet-6 | ResNetAP-10 | ResNet-18 |
| Random | $17.0_{\pm 0.3}$ | $19.1_{\pm 0.4}$ | $17.5_{\pm 0.5}$ | $24.8_{\pm 0.2}$ | $26.7_{\pm 0.5}$ | $25.5_{\pm 0.3}$ |
| Herding (Welling, 2009) | $17.2_{\pm 0.3}$ | $19.8_{\pm 0.3}$ | $16.1_{\pm 0.2}$ | $24.3_{\pm 0.4}$ | $27.6_{\pm 0.1}$ | $24.7_{\pm 0.1}$ |
| IDC-1 (Kim et al., 2022) | $\mathbf{24.3}_{\pm 0.5}$ | $25.7_{\pm 0.1}$ | $\mathbf{25.1}_{\pm 0.2}$ | $28.8_{\pm 0.3}$ | $29.9_{\pm 0.2}$ | $30.2_{\pm 0.2}$ |
| MinMaxDiff (Gu et al., 2024) | $22.3_{\pm 0.5}$ | $24.8_{\pm 0.2}$ | $22.5_{\pm 0.3}$ | $29.3_{\pm 0.4}$ | $32.3_{\pm 0.1}$ | $31.2_{\pm 0.1}$ |
| **MGD$^3$ (Ours)** | $23.4_{\pm 0.9}$ | $\mathbf{25.8}_{\pm 0.5}$ | $23.6_{\pm 0.4}$ | $\mathbf{30.6}_{\pm 0.4}$ | $\mathbf{33.9}_{\pm 1.1}$ | $\mathbf{32.6}_{\pm 0.4}$ |
| Full | $79.9_{\pm 0.4}$ | $80.3_{\pm 0.2}$ | $81.8_{\pm 0.7}$ | $79.9_{\pm 0.4}$ | $80.3_{\pm 0.2}$ | $81.8_{\pm 0.7}$ |

Table 2: Performance comparison on ImageNet-100. The best results are marked as **bold**.

**ImageNet-100 and ImageNet-1K.** Table 2 shows comparison to SOTA in ImageNet-100 in IPC 10 and 20 in various target architectures. Our method surpasses the previous SOTA by 1.3%, 1.6%, and 1.4% in IPC 20 for various target architectures. It also outperformed the MinMax diffusion approach in IPC 10 and achieved the best performance with the ResNetAP-10 target architecture while delivering the second-best results for ConvNet-6 and ResNet-18 architectures. It is important to note that our method is substantially more computationally efficient compared to IDC and MinMax (see Computational Cost below). We also compare our method with SOTA in ImageNet-1K on the soft-label protocol on IPC 10 and 50 in Table 4. Our method achieved SOTA outperforming previous SOTA by 1.3% and 1.6%. In the case of using a general-purpose diffusion in Imagenet-1k, our method shows an improvement of 3.4% and 2.3% in IPC 10 and IPC 50 over Stable Diffusion.

| Test Model | Mode Disc. | Mode Guid. | Stop Guid. | Acc. |
|---|---|---|---|---|
| ConvNet-6 | ✔ | - | - | $53.2_{\pm 1.4}$ |
| ResNetAP-10 | | | | $57.1_{\pm 1.3}$ |
| ResNet-18 | | | | $53.5_{\pm 0.6}$ |
| ConvNet-6 | ✔ | ✔ | - | $57.5_{\pm 1.3}$ |
| ResNetAP-10 | | | | $63.8_{\pm 1.6}$ |
| ResNet-18 | | | | $62.0_{\pm 2.2}$ |
| ConvNet-6 | ✔ | ✔ | ✔ | $\mathbf{59.6}_{\pm 2.2}$ |
| ResNetAP-10 | | | | $\mathbf{66.4}_{\pm 2.4}$ |
| ResNet-18 | | | | $\mathbf{64.4}_{\pm 1.9}$ |

Table 3: Ablation study on the component of our proposed method. The results are on the ImageNette dataset with IPC 10. Each component contributes to the overall performance.

| Method | IPC 10 | IPC 50 |
|---|---|---|
| SRe$^2$L (Yin et al., 2023) | $21.3_{\pm 0.6}$ | $46.8_{\pm 0.2}$ |
| RDED (Sun et al., 2024) | $42.0_{\pm 0.1}$ | $56.5_{\pm 0.1}$ |
| DiT (Peebles & Xie, 2023) | $39.6_{\pm 0.4}$ | $52.9_{\pm 0.6}$ |
| MinMax (Gu et al., 2024) | $44.3_{\pm 0.5}$ | $58.6_{\pm 0.3}$ |
| **MGD$^3$ (Ours)** | $\mathbf{45.6}_{\pm 0.1}$ | $\mathbf{60.2}_{\pm 0.1}$ |
| General Purpose | | |
| $D^4M$ (Su et al., 2024) | 27.9 | 55.2 |
| SD (Rombach et al., 2022) | $38.8_{\pm 0.2}$ | $56.2_{\pm 0.1}$ |
| SD + **MGD$^3$ (Ours)** | $42.2_{\pm 0.4}$ | $58.5_{\pm 0.2}$ |

Table 4: Overall accuracy comparison. Our method outperforms the current state-of-the-art on ImageNet-1k.

**Comparison versus Generative Prior Methods.** We compared our method with GLaD, H-GLaD, and LM3D on their cross-architecture setup, where AlexNet, VGG11, ResNet18, and ViT were

used for performance evaluation. The evaluation was done by running the evaluation five times per architecture and reporting the mean performance across all the architectures. We evaluated our model in 5 subsets: A, B, C, D, and E of ImageNet. Our method was trained using the hard-label protocol. Table 5 shows that our method outperformed previous methods in this setup. Also, it is worth noting that these methods are challenging to scale to large datasets such as Imagenet-1K or higher IPC (>50) due to higher time and space complexities.

**Computational Cost:** Our method achieves state-of-the-art performance on all datasets, except ImageNet-100, where the best-performing method, IDC-1 (Kim et al., 2022), has slightly better results than ours but with much higher computational cost. For example, MinMax (Gu et al., 2024) took 10 hours to produce a distilled dataset for ImageNet-100 with IPC-10, while IDC-1 (Kim et al., 2022) took over 100 hours for the same. The optimization strategy proposed in IDC-1 (Kim et al., 2022) can not scale up to the ImageNet-1K, and MinMax diffusion requires expensive fine-tuning of the diffusion model, especially for larger datasets like ImageNet-1k. In contrast, we used pre-trained diffusion models to create a distilled dataset with no additional computational cost for fine-tuning and minimal overhead for mode discovery. For comparison, our method takes 0.42 hours to generate a synthetic dataset for ImageNet-100 with IPC-10. This highlights the computational efficiency of our model compared to previous approaches.

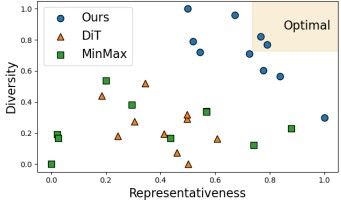

Figure 3: Representative score versus Diversity score for each class on Nette for IPC 10 versus various models.

| Distil Alg. | Method | ImNet-A | ImNet-B | ImNet-C | ImNet-D | ImNet-E | All |
|---|---|---|---|---|---|---|---|
| DC | Pixel | 52.3±0.7 | 45.1±8.3 | 40.1±7.6 | 36.1±0.4 | 38.1±0.4 | 42.3±3.5 |
|  | GLaD | 53.1±1.4 | 50.1±0.6 | 48.9±1.1 | 38.9±1.0 | 38.4±0.7 | 45.9±1.0 |
|  | H-GLaD | 54.1±1.2 | 52.0±1.1 | 49.5±0.8 | 39.8±0.7 | 40.1±0.7 | 47.1±0.9 |
|  | LM3D | 55.2±1.0 | 51.8±1.4 | 49.9±1.3 | 39.5±1.0 | 39.0±1.3 | 47.1±1.2 |
| DM | Pixel | 44.4±0.5 | 52.6±0.4 | 50.6±0.5 | 47.5±0.7 | 35.4±0.4 | 36.0±0.5 |
|  | GLaD | 52.8±1.0 | 51.3±0.6 | 49.7±0.4 | 36.4±0.4 | 38.6±0.7 | 45.8±0.6 |
|  | H-GLaD | 55.1±0.5 | 54.2±0.5 | 50.8±0.4 | 37.6±0.6 | 39.9±0.7 | 47.5±0.5 |
|  | LM3D | 57.0±1.3 | 52.3±1.1 | 48.2±4.9 | 39.5±1.5 | 39.4±1.8 | 47.3±2.1 |
| - | MGD$^3$ **(Ours)** | **63.4±0.8** | **66.3±1.1** | **58.6±1.2** | **46.8±0.8** | **51.1±1.0** | **57.2±1.0** |

Table 5: Comparison of our method with Generative Prior methods on ImageNet subsets A to E with IPC-10.

## 4.2 Ablation Study

**When should guidance stop?** To determine when to stop the guidance, we assessed mode guidance with $t_{SG}$ ranging from 50 to 0 in increments of 5 steps. A stop guidance of $t_{SG} = 50$ means no guidance, while $t_{SG} = 0$ means full guidance. Figure 4a shows that the optimal range to stop the guidance is between $t_{SG} = 30$ and $t_{SG} = 10$, with the peak at $t_{SG} = 20$. Additionally, Figure 4b illustrates that the guidance introduces more variability in the generation, with a more diverse set of backgrounds and poses. However, when the mode guidance is extended (e.g. $t_{SG} = 0$), it does not guarantee class consistency, as demonstrated in Figure 4b.

**Effect of each component.** To assess the impact of each proposed component, we incrementally evaluated the following: 1) Mode Discovery, 2) Mode Guidance, and 3) Stop Guidance. Mode Discovery involves performing $K$-means per class on the original dataset and selecting the closest sample to the k-means centroid. We conducted the evaluation on the ImageNette dataset with IPC 10, and reported the accuracy of ConvNet-6, ResNet10 with average pooling, and ResNet18. Table 3 demonstrates that using diffusion with mode guidance enhances mode discovery and that stop guidance is crucial for achieving improved performance.

**Visualzaling t-SNE.** To analyze the distilled dataset's coverage, we visualize a t-SNE plot of the distilled dataset from the DiT, MinMax Diffusion, and our method. Figure 5 illustrates that the DiT distilled dataset is mostly contained in one region of the original dataset distribution, while MinMax Diffusion extends to a broader area of the data distribution. However, the distilled dataset from our method covers a broader area of the data distribution than both methods.

**Representativeness versus Diversity.** While t-SNE provides a qualitative visualization of diversity, it does not present the complete picture. We are also interested in representativeness. With this in mind, our goal is to empirically measure diversity and representativeness in the t-SNE space described above. To measure diversity, we calculate the pairwise distance of all samples within a class for the

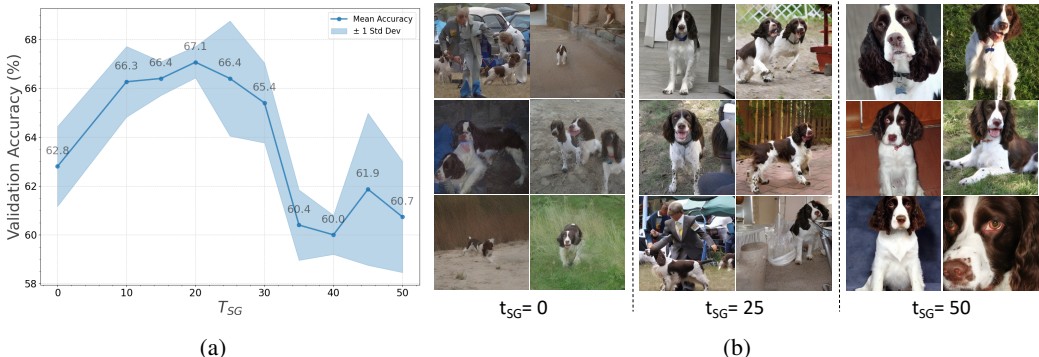

$t_{SG}= 0$      $t_{SG}= 25$      $t_{SG}= 50$

(a)          (b)

Figure 4: Ablation of the effect of $t_{SG}$, where $t_{SG} = 0$ denotes full guidance and $t_{SG} = 50$ denotes no guidance. (a) Shows validation accuracy versus $t_{SG}$ on ImageNette dataset. Best performance is achieved when $t_{SG}$ ranges between 20 and 30. (b) Shows generated images for the 'English Springer' class with full guidance ($t_{SG} = 0$), with early-stop guidance $t_{SG} = 25$ and no guidance ($t_{SG} = 50$). With early-stop guidance, the generated samples have more diversity w.r.t to the pose and background.

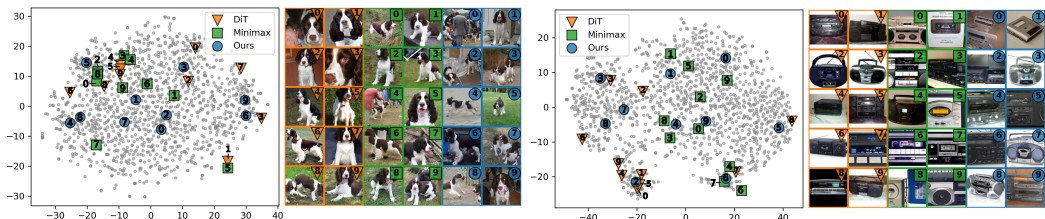

Figure 5: T-SNE plot showing the original samples (●) and the synthetic samples generated by different diffusion-based methods for two classes (English springer and cassette player) from ImageNet-1k. This visualization shows that DiT (Peebles & Xie, 2023) has limited diversity, Minmax (Gu et al., 2024) diffusion shows diversity but lacks full coverage, while our approach demonstrates mode diversity, achieving higher coverage.

distilled dataset and report the minimum distance per sample. To measure representativeness, we aim to assess how close a condensed sample is to the closest mode in the original dataset. To do this, we calculate the mean distance of the closest 50 examples in the original dataset. Note that greater mean distance means least representative and less mean distance means more representative.

We compared the diversity and representativeness of each class for DiT, MinMax diffusion, and our method as shown in Figure 3. Figure 3 presents normalized scores. We expressed high representativeness as a high value by visualizing a $1 - representativeness$ to aid visualization. Our experiment indicates that DiT examples show partial representative and partial diversity. On the other hand, MinMax produces more diverse examples than DiT, although some classes lack diversity. Our method demonstrates that our samples are both diverse and representative. Furthermore, we provide additional results about representativeness and diversity in the supplemental material.

## 5 CONCLUSION

Dataset distillation is an important task of condensing information from large training sets. Despite several efforts, the distilled datasets have limited representativeness and diversity in their synthetic samples. Our proposed method using latent diffusion with mode guidance overcame this limitation and demonstrated superior performance for the dataset distillation task across various benchmarks and setups. Specifically, our approach outperformed previous methods without the need for fine-tuning, as evidenced by our results on ImageNette, ImageIDC, ImageNet-100, and ImageNet-1K. We also analyzed and discussed the importance of different components of our method and showed their utility through rigorous ablation studies. We've shown that our approach can be used with general diffusion, like Text-to-Image Stable Diffusion, even when the training data doesn't overlap with the target dataset. In the future, we plan to extend our method to different applications where diverse representative samples are needed, such as continual learning.

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

## A  CLASS WISE DIVERSITY AND REPRESENTATIVENESS

Figure 6 shows the diversity and representativeness of each distilled sample for ten classes in the ImageNet-1k dataset for DiT, MinMax, and ours. This Figure shows that our method is consistent having higher representativeness across all the classes in comparison with the previous methods. Overall, our method maintains high diversity across most of the samples within a class. We observe that both MinMax and DiT consistently have a few samples with very low diversity.

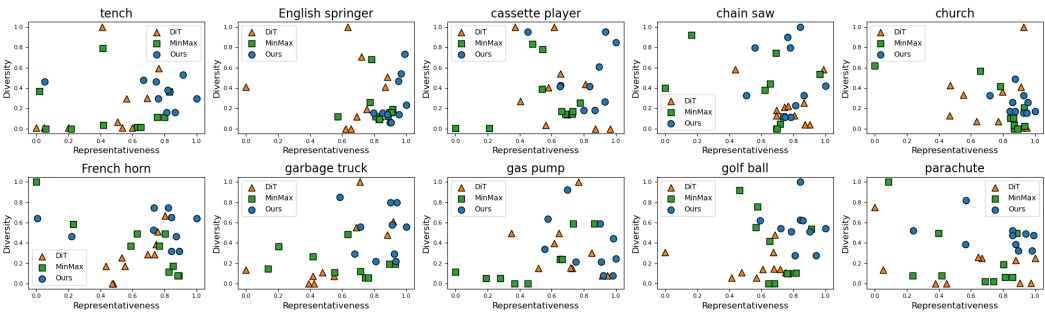

Figure 6: Representativeness versus Diversity by class for the distilled dataset from diffusion-based methods on 10 IPC of ImageNet-1k. Each point represents an image of the distilled dataset. DiT shows high representativeness but lacks diversity; MinMax shows diversity but lacks representativeness; Ours shows both diversity and representativeness.

## B  EFFECT OF STOP GUIDANCE IN DIVERSITY AND REPRESENTATIVENESS

In order to understand how the stop guidance affects the diversity and representation of the distilled dataset, we performed an evaluation of these metrics on the ImageNette dataset for IPC 10 for various $t_{SG}$ ranging from 50 to 0 our results are displayed on Figure 7. Our results show that when mode guidance is applied for any evaluated stop guidance $t_{SG}$, our method increases diversity. Our data shows that while stopping the mode guidance very early in the reverse process started from $t_{SG} = 45$ to $t_{SG} = 35$ mode guidance increases diversity. To our surprise, when the stop guidance is delayed toward the end of the reverse process from $t_{SG} = 30$ to $t_{SG} = 0$, the diversity starts to saturate, but the mode guidance increases the representativeness of the distilled dataset.

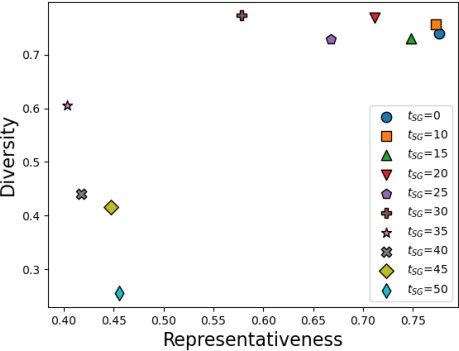

Figure 7: Representativeness versus Diversity versus $t_{SG}$. Each point represents a distilled dataset. Diversity and representativeness are obtained by computing the mean across all the samples in the distilled dataset. Stopping the mode guidance early in the reverse process ($t_{SG} = 45$ to $t_{SG} = 35$) promotes diversity. While prolonging the mode guidance between $t_{SG} = 35$ and $t_{SG} = 0$ increases representativeness.

## C  EFFECT OF MODE DISCOVERY ALGORITHM

To investigate the impact of the mode discovery algorithm, we assessed the following methods: random selection from the original dataset, k-means centroids, closest sample to k-means centroid, and DBSCAN. The evaluation is performed in ImageNette with IPC 10. For DBSCAN modes, we computed the mean of a cluster. We applied mode guidance with $t_{SG} = 25$ with estimated modes for each method. The results are presented in Table 6, showing that k-means centroids provided the best performance.

| Mode Discovery method | Accuracy |
|---|---|
| Random | $59.6_{\pm1.8}$ |
| DBSCAN | $61.3_{\pm1.9}$ |
| $k$-Means (closest sample) | $64.6_{\pm0.4}$ |
| **$k$-Means (centroid)** | $66.4_{\pm2.4}$ |

Table 6: Mode discovery algorithm versus Accuracy on ImageNette with IPC-10.

Table 7: Performance comparison with pre-trained diffusion models and other state-of-the-art methods on ImageWoof. All the results are reproduced by us on the $256\times256$ resolution. The missing results are due to out-of-memory. The best results are marked as **bold**. Higher is better. Results shown for the previous works are from Gu et al. (2024).

| IPC (Ratio) | Test Model | Random | Herding Welling (2009) | DiT Peebles & Xie (2023) | DM Zhao & Bilen (2023) | IDC-1 Kim et al. (2022) | GLaD Cazenavette et al. (2023) | MinMaxDiff Gu et al. (2024) | $MGD^3$ **(Ours)** | Full |
|---|---|---|---|---|---|---|---|---|---|---|
| 10 (0.8%) | ConvNet-6 | $24.3_{\pm1.1}$ | $26.7_{\pm0.5}$ | $34.2_{\pm1.1}$ | $26.9_{\pm1.2}$ | $33.3_{\pm1.1}$ | $33.8_{\pm0.9}$ | $\mathbf{37.0}_{\pm1.0}$ | $34.73_{\pm1.1}$ | $86.4_{\pm0.2}$ |
| | ResNetAP-10 | $29.4_{\pm0.8}$ | $32.0_{\pm0.3}$ | $34.7_{\pm0.5}$ | $30.3_{\pm1.2}$ | $39.1_{\pm0.5}$ | $32.9_{\pm0.9}$ | $39.2_{\pm1.3}$ | $\mathbf{40.4}_{\pm1.9}$ | $87.5_{\pm0.5}$ |
| | ResNet-18 | $27.7_{\pm0.9}$ | $30.2_{\pm1.2}$ | $34.7_{\pm0.4}$ | $33.4_{\pm0.7}$ | $37.3_{\pm0.2}$ | $31.7_{\pm0.8}$ | $37.6_{\pm0.9}$ | $\mathbf{38.5}_{\pm2.5}$ | $89.3_{\pm1.2}$ |
| 20 (1.6%) | ConvNet-6 | $29.1_{\pm0.7}$ | $29.5_{\pm0.3}$ | $36.1_{\pm0.8}$ | $29.9_{\pm1.0}$ | $35.5_{\pm0.8}$ | - | $37.6_{\pm0.2}$ | $\mathbf{39.0}_{\pm3.46}$ | $86.4_{\pm0.2}$ |
| | ResNetAP-10 | $32.7_{\pm0.4}$ | $34.9_{\pm0.1}$ | $41.1_{\pm0.8}$ | $35.2_{\pm0.6}$ | $43.4_{\pm0.3}$ | - | $\mathbf{45.8}_{\pm0.5}$ | $43.6_{\pm1.6}$ | $87.5_{\pm0.5}$ |
| | ResNet-18 | $29.7_{\pm0.5}$ | $32.2_{\pm0.6}$ | $40.5_{\pm0.5}$ | $29.8_{\pm1.7}$ | $38.6_{\pm0.2}$ | - | $\mathbf{42.5}_{\pm0.6}$ | $41.9_{\pm2.1}$ | $89.3_{\pm1.2}$ |
| 50 (3.8%) | ConvNet-6 | $41.3_{\pm0.6}$ | $40.3_{\pm0.7}$ | $46.5_{\pm0.8}$ | $44.4_{\pm1.0}$ | $43.9_{\pm1.2}$ | - | $53.9_{\pm0.6}$ | $\mathbf{54.5}_{\pm1.6}$ | $86.4_{\pm0.2}$ |
| | ResNetAP-10 | $47.2_{\pm1.3}$ | $49.1_{\pm0.7}$ | $49.3_{\pm0.2}$ | $47.1_{\pm1.1}$ | $48.3_{\pm1.0}$ | - | $56.3_{\pm1.0}$ | $\mathbf{56.5}_{\pm1.9}$ | $87.5_{\pm0.5}$ |
| | ResNet-18 | $47.9_{\pm1.8}$ | $48.3_{\pm1.2}$ | $50.1_{\pm0.5}$ | $46.2_{\pm0.6}$ | $48.3_{\pm0.8}$ | - | $57.1_{\pm0.6}$ | $\mathbf{58.3}_{\pm1.4}$ | $89.3_{\pm1.2}$ |
| 70 (5.4%) | ConvNet-6 | $46.3_{\pm0.6}$ | $46.2_{\pm0.6}$ | $50.1_{\pm1.2}$ | $47.5_{\pm0.8}$ | $48.9_{\pm0.7}$ | - | $\mathbf{55.7}_{\pm0.9}$ | $55.1_{\pm2.5}$ | $86.4_{\pm0.2}$ |
| | ResNetAP-10 | $50.8_{\pm0.6}$ | $53.4_{\pm1.4}$ | $54.3_{\pm0.9}$ | $51.7_{\pm0.8}$ | $52.8_{\pm1.8}$ | - | $58.3_{\pm0.2}$ | $\mathbf{60.2}_{\pm2.4}$ | $87.5_{\pm0.5}$ |
| | ResNet-18 | $52.1_{\pm1.0}$ | $49.7_{\pm0.8}$ | $51.5_{\pm1.0}$ | $51.9_{\pm0.8}$ | $51.1_{\pm1.7}$ | - | $58.8_{\pm0.7}$ | $\mathbf{59.7}_{\pm2.7}$ | $89.3_{\pm1.2}$ |
| 100 (7.7%) | ConvNet-6 | $52.2_{\pm0.4}$ | $54.4_{\pm1.1}$ | $53.4_{\pm0.3}$ | $55.0_{\pm1.3}$ | $53.2_{\pm0.9}$ | - | $\mathbf{61.1}_{\pm0.7}$ | $60.1_{\pm1.2}$ | $86.4_{\pm0.2}$ |
| | ResNetAP-10 | $59.4_{\pm1.0}$ | $61.7_{\pm0.9}$ | $58.3_{\pm0.8}$ | $56.4_{\pm0.8}$ | $56.1_{\pm0.9}$ | - | $64.5_{\pm0.2}$ | $\mathbf{66.5}_{\pm1.0}$ | $87.5_{\pm0.5}$ |
| | ResNet-18 | $61.5_{\pm1.3}$ | $59.3_{\pm0.7}$ | $58.9_{\pm1.3}$ | $60.2_{\pm1.0}$ | $58.3_{\pm1.2}$ | - | $65.7_{\pm0.4}$ | $\mathbf{68.8}_{\pm0.7}$ | $89.3_{\pm1.2}$ |

## D  EVALUATION ON IMAGEWOOF

**ImageWoof.** We compared our method with SOTA in ImageWoof on IPC 10, 20, 50, 70, and 100 on various target architectures, as shown in Table 7. It is worth noticing that this dataset is a fine-grained dataset where all classes belong to dog breeds. Due to its granularity of features, we trained DiT XL/2 on the ImageWoof dataset with just the simple loss mentioned in Eq. 2 following the same training epochs as (Gu et al., 2024). Our method outperformed the previous SOTA across various IPC values for different target architectures. Notably, our method demonstrated superior performance in all IPC values for the ResNet-18 architecture, achieved SOTA in IPC 10, 50, 70, and 100 with the ResNetAP-10 architecture, and delivered the best performance in IPC 20 and 50 with the ConvNet-6 architecture.

## E  EFFECT OF MODE GUIDANCE SCALE $\lambda$

To study how the mode guidance scale $\lambda$ affects performance, we evaluated the various values for $\lambda$ on ImageNette with IPC 10 with ResNetAP-10. Our results showed that when the mode guidance is too high, it's catastrophic for the distilled data, dropping the performance significantly; however, the best parameter was achieved by $\lambda = 0.1$.

## F  MODE GUIDANCE WITH DDIM

Our approach, similar to classifier guidance (Nichol & Dhariwal, 2021), can be incorporated guidance into DDIM using algorithm 1. In Table 8, we compare the effect of our approach in DDPM and

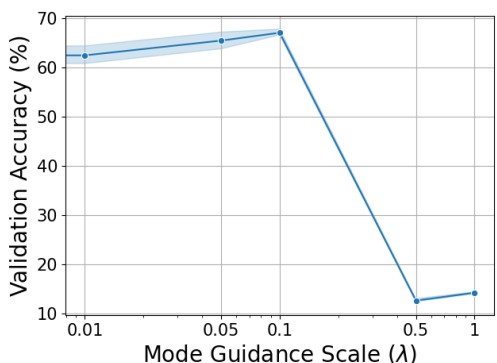

Figure 8: Effect of guidance scale on performance.

DDIM across LDM and DiT diffusion architectures. Our results demonstrate the effectiveness of our method with denoising samplers in both architectures, showcasing its flexibility with respect to diffusion architecture and sampler choice. This highlights the significant impact of our approach in enhancing the performance while being adaptable with different denoising diffusion models.

| Method | DDPM | DDIM |
|---|---|---|
| LDM | $60.3_{\pm 3.6}$ | $60.4_{\pm 3.1}$ |
| LDM + MGD³ (Ours) | $61.9_{\pm 4.1}$ | $62.3_{\pm 1.1}$ |
| DiT | $58.8_{\pm 2.1}$ | $61.4_{\pm 2.4}$ |
| DiT + MGD³ (Ours) | $66.4_{\pm 2.4}$ | $66.6_{\pm 0.6}$ |

Table 8: Comparison of performance between different diffusion models (LDM, DiT) with and without our approach, evaluated using DDPM and DDIM sampling methods.

---

**Algorithm 1** Mode Guidance with DDIM sampling, given a diffusion model $\epsilon_\theta(x_t)$, an estimated mode $m_k$ and mode guidance scale $\lambda$.

---

Input: estimated mode $m_k$ and mode guidance scale $\lambda$
$x_T \leftarrow$ sample from $\mathcal{N}(0, \mathbf{I})$
**for all** $t$ from $T$ to $1$ **do**
  $\mathbf{g_t} = (m_i - \hat{x}_0^t)$
  $\hat{\epsilon} \leftarrow \epsilon_\theta(x_t) - \sqrt{1 - \bar{\alpha}_t} \cdot \lambda \cdot \mathbf{g_t}$
  $x_{t-1} \leftarrow \sqrt{\bar{\alpha}_{t-1}} \left( \frac{x_t - \sqrt{1 - \bar{\alpha}_t}\hat{\epsilon}}{\sqrt{\bar{\alpha}_t}} \right) + \sqrt{1 - \bar{\alpha}_{t-1}}\hat{\epsilon}$
**end for**
**return** $x_0$

---

## G   DIVERSITY CLASS-WISE DIVERSITY SCORE

We calculated the diversity score for each class by averaging the diversity score across all the samples. Table 9 shows the diversity score for each class for DiT, MinMax, and Mode Guidance. Our method consistently generates a more diverse set for each class on ImageNette than the other methods.

## H   HARD-LABEL VERSUS SOFT-LABEL PROTOCOLS

We conducted further analysis on ImageNet-100, where we tested our approach from IPC-10 up to IPC-100. As illustrated in Table 10, our performance steadily improved, reaching $57.8_{\pm 0.2}$ with the hard-label protocol. Additionally, we compared the performance of ImageNet-100 using soft-label training on IPC-10, 20, 50, and 100. The results underscore a substantial performance boost when employing soft-labels.

| class | DiT | MinMax | Ours |
|---|---|---|---|
| tench | 0.35 | 0.18 | **0.82** |
| English springer | 0.65 | 0.33 | **0.62** |
| cassette player | 0.55 | 0.52 | **1.00** |
| chain saw | 0.00 | 0.37 | **0.55** |
| church | 0.54 | 0.41 | **0.77** |
| French horn | 0.21 | 0.13 | **0.54** |
| garbage truck | 0.44 | 0.38 | **0.76** |
| gas pump | 0.50 | 0.24 | **0.67** |
| golf ball | 0.20 | 0.33 | **0.78** |
| parachute | 0.08 | 0.48 | **0.79** |
| Average | 0.35 | 0.34 | **0.73** |

Table 9: Results: Comparison of per-class diversity scores on ImageNette with IPC-10

| Method | Labels | IPC10 | IPC20 | IPC50 | IPC 100 |
|---|---|---|---|---|---|
| **MGD$^3$ (Ours)** | Hard-Label | $23.6_{\pm 0.4}$ | $32.6_{\pm 0.4}$ | $51.8_{\pm 0.2}$ | $57.8_{\pm 0.2}$ |
| | Soft-label | $34.0_{\pm 1.0}$ | $50.2_{\pm 0.7}$ | $69.2_{\pm 0.4}$ | $75.8_{\pm 0.3}$ |

Table 10: Evaluation of training with hard-labels versus soft labels in ImageNet-100 training with ResNet18.

# I    VISUALIZATION OF DENOISING TRAJECTORIES WITH MODE GUIDANCE FOR DIFFERENT $t_{SG}$

See Figure 9

# J    EVALUATION TECHNICAL DETAILS

For the hard-label protocol, we followed the evaluation method described in (Gu et al., 2024). We trained our model on a synthetic dataset for 1500 epochs for IPC values of 20, 50, and 100, and extended the training to 2000 epochs for an IPC value of 10. We used Stochastic Gradient Descent (SGD) as the optimizer, setting the learning rate at 0.01. We used a learning rate decay scheduler at the 2/3 and 5/6 points of the training process, with the decay factor (gamma) set to 0.2. Cross-entropy was used as the Loss objective.

For the soft-label protocol, we followed the evaluation used by (Gu et al., 2024; Sun et al., 2024) for ImageNet-1k evaluation. We evaluate the model by training a network for 300 epochs with Resnet-18 architecture as both teacher and student. We used the AdamW optimizer, with a learning rate set at 0.001, a weight decay of 0.01, and the parameters $\beta_1 = 0.9$ and $\beta_2 = 0.999$.

918
919
920
921
922
923
924
925
926
927
928
929
930
931
932
933
934
935
936
937
938
939
940
941
942
943
944
945
946
947
948
949
950
951
952
953
954
955
956
957
958
959
960
961
962
963
964
965
966
967
968
969
970
971

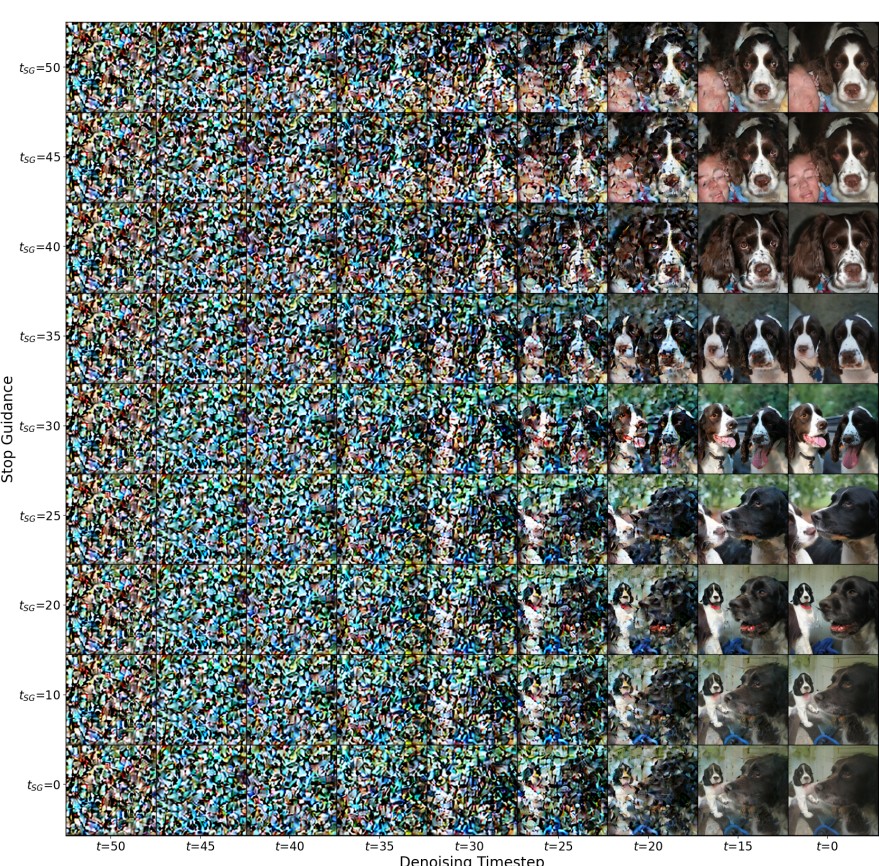

Figure 9: Generated images through the reverse process for different values of $T_{SG}$.

