# OpenReview forum: "MGD$^3$: Mode-Guided Dataset Distillation using Diffusion Models"
_ICLR.cc/2025/Conference — Submitted to ICLR 2025_

### Official Review · Reviewer_LHFV · 2024-10-26

**Soundness:** 3
**Presentation:** 3
**Contribution:** 3
**Rating:** 8
**Confidence:** 4

**Summary:**

This paper proposes a mode guidance method in the reverse process of a pretrained diffusion model to enhance the diversity of synthetic images generated for dataset distillation task. They calculate the mode by kmeans clustering on the original dataset, then calculate the mode guidance score which is added to the noise function at appropriate time steps during the reverse process of the diffusion model. This approach achieves both the representativeness and diversity in the synthetic images.

**Strengths:**

1. The quantitative results on ImageNet-1k and its subsets show improvement across all tables compared to both the baseline and state-of-the-art methods at various ipc.
2. The pretrained diffusion model generates distinct samples that ensure intra-class diversity with the help of the guidance signal in reverse process.
3. Training time is reduced as the pretrained diffusion model does not require fine-tuning.

**Weaknesses:**

1. The image quality, the information loss and the recovery of data distribution rely on the diffusion model heavily.
2. The mode simply calculated by kmeans may represent the original data distribution insufficiently and the author does not compare it with other cluster methods.

**Questions:**

1. Can you please tell if the results of CIFAR10/100 and tinyImageNet have also improved?
2. Have you tried other cluster methods in experiments to check if they influence performance?

---

> ### Author Response · Authors · 2024-11-22
>
> Thank you for spending your valuable time to provide us with your reviews. We tried to address your concerns in the rebuttal. We believe your feedback on our rebuttal is crucial. We would greatly appreciate if you can provide us with the feedback before the end of the discussion period.
>
> # W1: Diffusion Model requirements
> Our work primarily focuses on generating a distilled dataset given a pretrained model in the target distribution, assuming a well-trained diffusion model. Within this scope, we achieved state-of-the-art performance. While we also evaluated our method in an extreme case where the diffusion model was trained in a different dataset from the target distribution and demonstrated improvements over the baseline without requiring fine-tuning, this scenario lies outside the main focus of our study.
>
> We acknowledge that this limitation presents an interesting direction for future research. Specifically, investigating dataset distillation when the diffusion model does not adequately capture the target dataset's characteristics, potentially through fine-tuning, could further expand the applicability and robustness of this approach.
>
> # W2 and Q2. Experiments with other mode discovery algorithms
>
> We included this experiment in the Supplemental Material in Table 6 in the Original Submission.
>
> | Mode Discovery Method         | Accuracy        |
> |-------------------------------|-----------------|
> | Random                        | 59.6 ± 1.8      |
> | DBSCAN                        | 61.3 ± 1.9      |
> | $k$-Means (closest sample)    | 64.6 ± 0.4      |
> | **$k$-Means (centroid)**      | **66.4 ± 2.4**  |
> *Table: Mode discovery algorithm versus Accuracy on ImageNette with IPC-10.*
>
>
> # Q1. Extra Results
>
>
> We evaluated Stable Diffusion (SD), Stable Diffusion with Disentangled Diffusion, and Stable Diffusion with our proposed method using the soft-label protocol under IPC=10. This setup is particularly challenging because Stable Diffusion was not trained on the target distribution. However, if the diffusion model were trained on the target dataset, such as CIFAR-10 or CIFAR-100, our method would likely achieve even better performance as previous experiment showed.
>
> CIFAR10 - Text2Image
>
> | Model                       | Acc              |
> | --------------------------- | ---------------- |
> | SD                          | $42.37 \pm 1.69$ |
> | SD + Disentangled Diffusion | $44.03 \pm 0.74$ |
> | SD + Ours                   | $46.02 \pm 1.64$ |
>
> CIFAR100 - Text2Image
>
> | Model                       | Acc              |
> | --------------------------- | ---------------- |
> | SD                          | $44.20 \pm 0.81$ |
> | SD + Disentangled Diffusion | $46.62 \pm 0.48$ |
> | SD+ Ours                    | $46.46 \pm 0.34$ |

---

> > ### Comment · Reviewer_LHFV · 2024-11-26
> >
> > Thank you for the detailed response. I will keep the score.

---

### Official Review · Reviewer_rbyc · 2024-10-29

**Soundness:** 2
**Presentation:** 3
**Contribution:** 3
**Rating:** 6
**Confidence:** 4

**Summary:**

This paper proposes a mode-guided diffusion model that generates synthetic datasets using pre-trained models without the need for fine-tuning. The method operates through three key stages: mode discovery, mode guidance, and stop guidance. These stages ensure both enhanced data diversity and high sample quality. Furthermore, the approach is versatile, making it applicable to various diffusion models. Experimental results demonstrate that the proposed method surpasses existing techniques across multiple datasets, effectively addressing the issue of limited mode diversity in generated data.

**Strengths:**

- Ensuring dataset sample diversity has long been a challenge in the field of dataset distillation. This paper addresses this by employing mode guidance to generate as diverse samples as possible for each class, minimizing redundancy and significantly enhancing intra-class diversity in the generated dataset.

- The paper utilizes pre-trained diffusion models to generate datasets without the need for additional fine-tuning, relying only on guidance during the denoising process, which simplifies the approach and improves efficiency.

- The stop guidance mechanism strikes a balance between sample diversity and quality, ensuring diverse samples while maintaining high quality, and preventing potential negative effects from excessive guidance.

**Weaknesses:**

- The mechanisms of mode and stop guidance are clear but lack strong theoretical support.

- The method is easy to understand but could be optimized, such as by automating the adjustment of $t_{SG}$ and improving the mode discovery algorithm.

- The approach is still fundamentally about image generation via diffusion models, with insufficient exploration of its contribution to dataset distillation.

**Questions:**

- Could you go beyond the traditional framework of diffusion model-based image generation and further explain the relationship between your method and dataset distillation?

- I’m curious about the resource consumption of $MGD^3$, such as runtime and memory usage. Could you provide more details on this?

- How do you ensure that the modes obtained using the K-means clustering algorithm are meaningful in the context of dataset distillation? Have you tried other algorithms for mode discovery in each class?

- Given the increase in sample diversity, I would like to see a comparison of cross-architecture evaluations with other diffusion + DD methods. Could you include more network architectures? The paper only uses three networks for evaluation.


If you can address all or most of my questions, I would consider giving a higher score.

---

> ### Author Response · Authors · 2024-11-23
>
> Thank you for spending your valuable time to provide us with your reviews. We tried to address your concerns in the rebuttal. We believe your feedback on our rebuttal is crucial. We would greatly appreciate if you can provide us with the feedback before the end of the discussion period.
> # W1.1 Mode Guidance Theoretical Support
>
> Our mode guidance follows the classifier guidance formula,
> $$
> \hat{\epsilon_\theta}(x_t, t, c) = \epsilon_\theta(x_t, t, c) + \lambda \cdot \mathbf{g}_t \cdot \sigma_t
> $$
> where $\mathbf{g_t}$ is the vector towards the specific mode for this sample.  This guidance push the denoising process to ensure sampling from targeted regions (modes) of the data distribution. This aligns with support similar to that of guided diffusion [1].
>
> [1] Dhariwal, Prafulla, and Alexander Nichol. "Diffusion models beat gans on image synthesis." NeurIPS (2021).
>
>
> # W1.2 Stop Guidance Theoretical Support
> The rationale for stop guidance is informed by the stages of diffusion [2], which identify the refinement phase as critical for preserving fidelity. Strong mode guidance during this stage can leading to artifacts that can disrupt the semantic and quality consistency of the samples. By ceasing guidance at a predefined timestep $t_{SG}$ at 50% of the denoising process​ to ensure that the quality of the generation,  which is supported by empirical observations (e.g., Figure 4b).
>
>
> # W2 Suggested Future Work
>
> While automating stop guidance is an interesting direction for future work, we opted to stop at 50% of the generation process based on insights from [2], and our experiments demonstrated that this approach was effective in achieving state-of-the-art performance. Regarding mode discovery, further improvements may not be critical, as the diffusion model inherently pushes samples toward the most likely mode in the learned distribution of the class. As long as the estimated mode is sufficiently close to the true mode, the diffusion process ensures convergence to the actual class mode, making the current mode discovery approach effective.
>
>   [2] Yu, Jiwen, et al. "Freedom: Training-free energy-guided conditional diffusion model." CVPR 2023.
>
> # W3. Contribution to Dataset Distillation
>
> In this work, we presented a diffusion model that synthesizes a distilled efficiently without any fine-tuning. Our contribution focuses on exploiting learned distribution by pre-trained models to generate a distilled dataset efficiently without finetuning that scales to high-resolution images.
>
> # Q1. Relationship with dataset distillation
>
> Dataset distillation aims to condense a small dataset with a similar characteristic as the original dataset. One way to achieve this is to cover as many characteristics as possible with a minimal set of samples. To achieve this, we are enforcing by sampling each sample in the class from a different mode. Each mode captures a distinct class characteristic, which can help cover as many characteristics as possible with fewer samples. In a similar line of work, DREAM[3] uses a similar motivation with a different formulation in the non-generative dataset distillation approaches. Although we have similar motivations, our approach aims to guide the generation toward this mode with our mode guidance, which results in a method that scales to high-resolution images while being efficient.
>
> [3] Liu, Yanqing, et al. "Dream: Efficient dataset distillation by representative matching." CVPR. 2023.
>
> # Q2. Computational and Memory Usage
>
> Computational Cost was included in the Supplemental Material.
>
> For ImageNet-100 with IPC-10, it requires only **0.42 hours**, compared to **10 hours** for MinMax and over **100 hours** for IDC-1. Unlike IDC-1, which struggles to scale to larger datasets like ImageNet-1K, and MinMax, which demands expensive fine-tuning, our approach uses pre-trained diffusion models without additional fine-tuning, ensuring minimal computational overhead. In terms of GPU memory our method with DiT use 3.9GB at inference. While, MinMax Diffusion uses up to 16GB, due to the diffusion fine-tuning.

---

> > ### Author Response · Authors · 2024-11-23
> >
> > # Q3. Clarification Mode Discovery
> >
> > Our method uses the same mode discovery for each class. Take in mind that the mode guidance pushes toward the guided mode, however, the diffusion will push toward the more likely sample in the learned distribution of the class. If the estimated mode is close enough the diffusion will push toward the actual class mode. However, if the estimated mode is far from the actual mode is likely to affect the performance.
> >
> > We also included this experiment evaluating various Mode Discovery Methods in the Supplemental Material in Table 6 in the Original Submission.
> >
> > | Mode Discovery Method         | Accuracy        |
> > |-------------------------------|-----------------|
> > | Random                        | 59.6 ± 1.8      |
> > | DBSCAN                        | 61.3 ± 1.9      |
> > | $k$-Means (closest sample)    | 64.6 ± 0.4      |
> > | **$k$-Means (centroid)**      | **66.4 ± 2.4**  |
> > *Table: Mode discovery algorithm versus Accuracy on ImageNette with IPC-10.*
> > # Q4. Comparison with additional DD and Cross-Architectures
> >
> >   Comparison versus Generative Prior Methods Section we compared with others Dataset Distillation Methods on IPC 10 for ImageNet A-B on AlexNet, VGG11, ResNet18, and ViT architectures which are different to the ConvNet-6 and ResNetAP-10 on the other evaluations. Table 5 reports the mean across all the architectures.
> >
> > | Distil Alg. | Method         | ImNet-A       | ImNet-B       | ImNet-C       | ImNet-D       | ImNet-E       | All           |
> > |-------------|----------------|---------------|---------------|---------------|---------------|---------------|---------------|
> > | DC          | Pixel          | 52.3±0.7      | 45.1±8.3      | 40.1±7.6      | 36.1±0.4      | 38.1±0.4      | 42.3±3.5      |
> > |             | GLaD           | 53.1±1.4      | 50.1±0.6      | 48.9±1.1      | 38.9±1.0      | 38.4±0.7      | 45.9±1.0      |
> > |             | H-GLaD         | 54.1±1.2      | 52.0±1.1      | 49.5±0.8      | 39.8±0.7      | 40.1±0.7      | 47.1±0.9      |
> > |             | LM3D           | 55.2±1.0      | 51.8±1.4      | 49.9±1.3      | 39.5±1.0      | 39.0±1.3      | 47.1±1.2      |
> > | DM          | Pixel          | 44.4±0.5      | 52.6±0.4      | 50.6±0.5      | 47.5±0.7      | 35.4±0.4      | 36.0±0.5      |
> > |             | GLaD           | 52.8±1.0      | 51.3±0.6      | 49.7±0.4      | 36.4±0.4      | 38.6±0.7      | 45.8±0.6      |
> > |             | H-GLaD         | 55.1±0.5      | 54.2±0.5      | 50.8±0.4      | 37.6±0.6      | 39.9±0.7      | 47.5±0.5      |
> > |             | LM3D           | 57.0±1.3      | 52.3±1.1      | 48.2±4.9      | 39.5±1.5      | 39.4±1.8      | 47.3±2.1      |
> > | -           | **MGD³ (Ours)**| **63.4±0.8**  | **66.3±1.1**  | **58.6±1.2**  | **46.8±0.8**  | **51.1±1.0**  | **57.2±1.0**  |

---

> > > ### Comment · Reviewer_rbyc · 2024-11-24
> > >
> > > Thank you to the author for the detailed response. It has addressed most of my concerns, and I will increase my score to 6.

---

### Official Review · Reviewer_iaR4 · 2024-11-01

**Soundness:** 2
**Presentation:** 2
**Contribution:** 2
**Rating:** 3
**Confidence:** 4

**Summary:**

This paper adapts the pretrained diffusion model for dataset distillation. Specifically, with a pretrained diffusion model, the authors introduce mode guidance in the early denoising stages. The modes for guidance are discovered from the target data distribution with VAE encoder. Experiments show that the new method outperforms some diffusion model baselines and dataset distillation methods.

**Strengths:**

1.	The idea of mode guidance for target-distribution synthesis is reasonable and easy to implement.

2.	Extensive experiments and study have been provided.

**Weaknesses:**

1.	The writing can be improved, especially the logic and causality in introduction. The citation format is unsuitable. Some words/phrases have inconsistent capitalization.
2.	Some statements are not rigorous:
a)	The authors claim “… diffusion models … do not suffer from mode collapse” in introduction. Is it theoretically or empirically proved in previous work? It also has conflicts with the claims in other paragraphs.
b)	“We validate this by addressing the following question: Given a pre-trained diffusion model, can a distilled dataset be extracted from this model, as it has learned the data distribution?” As claimed in the abstract, the diffusion model is pre-trained by others on some datasets. It is not guaranteed that the diffusion model “has learned the data distribution” of “a distilled dataset”.
3.	The listed five “contributions” are mostly repeated and trivial.
4.	According to Table 1, the new method is better than some diffusion baselines, while obviously worse than the classic dataset distillation methods, e.g., DM 2023.
5.	Though images from target distribution are synthesized by diffusion model, there is no connection to dataset distillation, in which data/knowledge should be distilled/condensed.

**Questions:**

No.

---

> ### Author Response · Authors · 2024-11-22
>
> Thank you for spending your valuable time to provide us with your reviews. We tried to address your concerns in the rebuttal. We believe your feedback on our rebuttal is crucial. We would greatly appreciate if you can provide us with the feedback before the end of the discussion period.
>
> # W1 Minor Fixes
>
> We fixed the suggested changes this in the latest version, which is also uploaded to OpenReview.
>
> # W2.1 Clarification Diffusion and Mode Coverage
>
>
> Our statement is based on findings from prior work, such as [1], which explored the Generative Learning Trilemma. This study showed that diffusion models can achieve both high-quality outputs and diverse mode coverage, unlike GANs, which often sacrifice diversity for faster sampling. While these findings indicate that diffusion models are less likely to experience mode collapse, they are based on empirical evidence, not theoretical proof.
>
> [1] Xiao, Zhisheng, Karsten Kreis, and Arash Vahdat. "Tackling the Generative Learning Trilemma with Denoising Diffusion GANs." ICLR 2022.
>
> # W2.2 Clarification of our evaluated Settings
>
> Our focus on this work is using diffusion models to generate a distilled dataset. This can  be  done in 2 settings.
>
> In setting 1, using a pre-trained diffusion on the target dataset, can we generate a distilled dataset that is both representative and diverse? Our motivation for this is that diffusion is capable of providing mode coverage of the target distribution. However, even though the diffusion learns the target distribution, is prone to oversample from the more likely modes, especially generating a dataset with less diverse samples especially on low IPC settings (e.g., IPC=10). Our method aims to appropriately utilize  the pre-trained diffusion model to guarantee a diverse dataset generation. We showed that our method produced diverse distilled datasets in Figures 3, 5, and Table 9, which led to better performance.
>
> In setting 2, we also evaluated our method on diffusion that was trained on other distributions as the target distribution (for instance, text-to-image diffusion). This setting is extremely challenging, due to possible domain gaps between the learned and target distribution. We showed, empirically, that even though these methods perform worse, our method improves, the respective baselines showing that our method has the potential better to align the learned distribution to the target distribution even without finetuning.
>
> # W3 Comments about contributions
>
> To summarize our contributions and results more concisely.
>
> 1. We present a novel approach for dataset distillation by leveraging a pre-trained diffusion model to generate a distilled dataset.
> 2. We introduce Mode Guidance with Stop Guidance to effectively control the denoising process, which enhances intra-class diversity and improves class representativeness, as demonstrated in Figures 3 and 5.
> 3. Our method showcases improved or comparable performance in relation to the current state-of-the-art, without necessitating fine-tuning of the diffusion model. Notably, our approach achieves competitive outcomes even when employing a text-to-image model like Stable Diffusion, underscoring its adaptability and effectiveness across various diffusion models.

---

> > ### Author Response · Authors · 2024-11-22
> >
> > # W4 Clarification Table 1
> >
> > To clarify, our method is evaluated in two distinct scenarios:
> >
> > 1. **Diffusion Trained on the Target Distribution (Setting 1):**
> >     The primary focus of our work is on scenarios where the diffusion model is pre-trained on the target distribution. In this case, our method consistently outperforms all prior methods, including classic dataset distillation approaches such as DM 2023, which were also applied to models trained on the target distribution. This highlights the effectiveness of our approach in leveraging pre-trained diffusion models for the target dataset.
> >
> > 2. **General-Purpose Diffusion (Setting 2):**
> >     We also evaluate our method in a more challenging setting, where the diffusion model (e.g., Stable Diffusion) is trained on a broader dataset (LAION-2B) rather than the target distribution. While our method does not surpass DM 2023 in this scenario, it is important to note that DM 2023 is directly applied to the target dataset. **For a fair comparison, DM 2023 would need to be trained on LAION-2B and then evaluated on the target dataset, such as Nette.** The purpose of this evaluation is to demonstrate that our method can align a pre-trained diffusion model’s learned distribution to the target distribution, even when the model is not explicitly trained on it.
> >
> >
> > This distinction highlights the versatility of our method in both conventional and more general-purpose settings, offering unique advantages in cases where diffusion models are not directly trained on the target dataset.
> >
> > # W5 Relationship with Dataset Distillation
> >
> > Dataset distillation aims to condense a small dataset with a similar characteristic as the original dataset. One way to achieve this is to cover as many characteristics as possible with a minimal set of samples. To achieve this, we are enforcing by sampling each sample in the class from a different mode. Each mode captures a distinct class characteristic, which can help cover as many characteristics as possible with fewer samples. In a similar line of work, DREAM[1] uses a similar motivation with a different formulation in the non-generative dataset distillation approaches. Although we have similar motivations, our approach aims to guide the generation toward this mode with our mode guidance, which results in a method that scales to high-resolution images while being efficient.
> >
> > [1] Liu, Yanqing, et al. "Dream: Efficient dataset distillation by representative matching." CVPR. 2023.

---

> > ### Comment · Reviewer_iaR4 · 2024-11-25
> >
> > Thanks for the response! I am glad to see that the authors polished their submission based on the comments from reviewers. But I still think the submission is not ready to publish due to the unsuitable statements and the unconvincing experiments.

---

> > > ### Author Response · Authors · 2024-11-26
> > >
> > > We thank the reviewer for acknowledging the refinements made in our submission.
> > >
> > > Regarding the statements in the introduction, we would like to highlight that we have further revised and refined them in the latest version, taking into account the concerns raised. While we previously clarified the rationale behind these statements in our earlier response, we have now updated the statements (89-97) to be more suitable. If there are any remaining statements that require further attention, we would greatly appreciate specific guidance and will gladly address them in detail.
> > >
> > > Concerning Table 1 and the experimental setup, we previously explained the two evaluation settings: (1) a diffusion model trained on the target dataset and (2) a general-purpose diffusion model (e.g., Stable Diffusion) trained on a broader dataset. We are keen to better understand why these experiments are perceived as unconvincing and would greatly value further clarification from the reviewer. This will help us better address any concerns and improve the clarity and impact of our work.

---

> > > > ### Author Response · Authors · 2024-11-29
> > > >
> > > > We deeply appreciate the time and thoughtful effort you have invested in reviewing our manuscript. We have carefully addressed your comments in our rebuttal and are committed to resolving any remaining concerns. We kindly request your timely feedback to ensure we can provide comprehensive clarifications before the deadline.

---

> > > ### Author Response · Authors · 2024-12-02
> > >
> > > Thank you for all the suggestions and feedback. We have diligently taken up the suggestions and carefully updated the pdf. Also, we answered the queries in as much detail as possible. As the deadline for author discussion is approaching close, we would be happy to clarify if there are any further questions or concerns. Thank you !

---

### Official Review · Reviewer_JiWQ · 2024-11-04

**Soundness:** 2
**Presentation:** 2
**Contribution:** 1
**Rating:** 3
**Confidence:** 4

**Summary:**

This paper proposed a generative prior methods called MGD for dataset distillation tasks. This paper follows the idea of the baseline MiniMaxDiff Gu et al. (2024) to enhance the diversity of the generated synthetic data. MGD influences the generation process by leveraging DiT models without requiring re-training or fine-tuning. Specifically, the authors propove to utilize the so-called mode to regularize the generation of synthetic data. However, the formulation for calculating these modes is not clearly explained. Based on Figure 2, my understanding is that the modes are the cluster centers of the original dataset. Additionally, the paper lacks theoretical proof to support the effectiveness of the proposed mode-guided approach. The demonstration in Figure 1 selects only four data instances to illustrate diversity improvement, which seems highly subjective.

**Strengths:**

1. The authors consider the soft-label protocol and hard-label protocol, which are important for fair comparison in dataset distillation. We can see the difference between two the protocols in Table 10.

2. The performance improvements reported in this paper are significant.

**Weaknesses:**

1. It is good that the authors clarify the the soft-label protocol and hard-label protocol. However, what protocol is used in Table 1, 2, 3? If the soft-label protocol is used, what are the parameters of valiation epochs, teacher networks, and the data augmentation methods?

2. I do not see a clear formulation explaining the derivation of the modes. Only Section 3.2 (lines 307 to 319) discusses the application of modes. My main concern is with the method used to select the modes; specifically, specially designed parameters for generating these modes could significantly impact the final performance.

3. Based on Figure 2, my understanding is that the modes are the cluster centres of the original dataset. This approach is very close to the Dream method. Please tell more details about the difference between the two methods.


4. The abstract could be more concise. For example, the limitation mentioned from lines 12 to 20 can be condensed into two sentences.  The description of your proposed method could be more generalized. Additionally, the caption of Figure 1, and the related works could be more concise as well.

5. The authors should emphasize more about the derivation, and formulation on the modes. However, most of the introduction and related work are telling the basic information of the previous methods. I cannot find a strong motivation or intuition to develop the mode-guided approach. Also, i doubt the performance enhancement is highly affected by the parameters in obtaining the modes.

6. The random method achieves a very high performance as stated in Table 1. What if the set is constructed by dataset pruning method, such as [2], and is evaluated by the soft-label protocols?

7. Although the authors list five contributions in this paper, points 1, 3, 4, and 5 seem redundant, essentially representing the same idea. Additionally, point 2 merely reiterates a widely acknowledged finding from previous works—that diversity is beneficial.

[1] Liu, Yanqing, et al. "Dream: Efficient dataset distillation by representative matching." Proceedings of the IEEE/CVF International Conference on Computer Vision. 2023.

[2] He, M., Yang, S., Huang, T., & Zhao, B. (2024). Large-scale dataset pruning with dynamic uncertainty. In Proceedings of the IEEE/CVF Conference on Computer Vision and Pattern Recognition (pp. 7713-7722).

**Questions:**

As stated in weakness part.

---

> ### Author Response · Authors · 2024-11-22
>
> Thank you for spending your valuable time to provide us with your reviews. We tried to address your concerns in the rebuttal. We believe your feedback on our rebuttal is crucial. We would greatly appreciate if you can provide us with the feedback before the end of the discussion period.
>
> # W1 - Clarification of Soft-Label and Hard-Label
> In response to your concerns, Tables 1, 2, and 3 illustrate our use of the Hard-Label Protocol, as mentioned in the last sentence of paragraphs 345-350. For the soft-label protocol (used in Table 4), 1) we used a teacher network of ResNet-18, 2) We trained for 300 epochs, and 3) we used the same augmentation as Hard-Label (RandomCropResize and CutMix). We are including more detail in the supplemental material for both protocols, and we will make the code available for all the experiments upon acceptance.
>
> # W2.1 - Derivation of Modes
>
> Our goal in generating a diverse and representative distilled dataset is to ensure that it captures the key characteristics of the original dataset. Modes have the following characteristics:
>
> 1. **Representative instances of the class**: Each mode effectively summarizes a unique subset of the class distribution.
> 2. **Distinct instances within the class**: Any two modes represent significantly different characteristics or patterns within the same class, ensuring minimal redundancy.
>
> By sampling from different modes, we ensure diversity in the distilled dataset, as each mode contributes unique and complementary information. In our work, we ensure diversity by guiding each sample in the same class to different modes and hence bringing diversity across generated samples of the distilled dataset. To achieve this goal, we enforce the diffusion model by adding mode guidance in the denoising process.
> Specifically, the denoising process  is guided as follows (Equation 6 Main paper):
>
> $$
> \hat{\epsilon_\theta}(x_t, t, c) = \epsilon_\theta(x_t, t, c) + \lambda \cdot \mathbf{g}_t \cdot \sigma_t
> $$
>
> where, $\mathbf{g_t}$ is a vector towards the specific mode $m_k$ for this sample. Note that a pre-trained model is used without finetuning with a dataset distillation objective. Mode guidance is applied only at inference.
> # W2.2 - Details about Mode Discovery
>
> For mode selection, we experimented with several methods (Table 6) we found that K-Means was a simple yet effective Mode Discovery approach. We intuitively set the number of clusters (K) to the number of images per class (IPC) so that each sample can have as distinct characteristics as possible. **There are no additional parameters involved in the clustering on the K-means additional to the K**.  Despite its simplicity, this single-parameter method demonstrated robust performance with our mode guidance.
> The supplemental material further explores how various mode discovery methods impact final performance.
>
> # W3 Comparison with DREAM
> In the DREAM paper, the data is divided into clusters, and then each cluster is separately condensed using an optimization-based condensation method (Gradient Matching or Distribution Matching) to ensure sample diversity. In contrast, our method uses a generative model-based condensation approach. While DREAM can be plugged into these generative methods, it would require training a diffusion model conditioned on both the class and the mode. In contrast, our proposed method leverages a pre-trained diffusion to guide the denoising process toward certain modes using our mode guidance objective. This novel idea of mode guidance helps our approach achieve state-of-the-art performance without the need to train/finetune the diffusion model. In addition to mode guidance, we also propose early stopping in our approach, which provides control over the denoising trajectory. This early stopping provides an additional improvement of up to 4% on top of mode guidance (See Table 3 in the main paper). In terms of performance on ImageNet-1k, DREAM reported a score of 18.5 using a convolutional network with IPC-10. We evaluated our method using the same setup, and have achieved a significantly higher accuracy(i.e., 34.2 %).
>
> # W4 - Comment About Text
>
> Taking the suggested changes into consideration, we have  appropriately modified  the abstract, Figure 1, and the related works in the latest version of the paper.  Regarding the proposed method, we have ensured that it is generalized for any diffusion process. While our primary focus is on latent diffusion, the notation we use is general for any diffusion. We appreciate your insights and hope these adjustments enhance the clarity and conciseness of our work. Thank you again for your suggestions!

---

> > ### Author Response · Authors · 2024-11-22
> >
> > # W5.1 - Clarification about the Motivation
> >
> > The MinMax Diffusion approach, while effective at fine-tuning models for representativeness and diversity, lacks explicit control during the sampling process. This absence of control often results in inconsistencies in sample diversity, as shown in Figure 5 of the main paper.
> >
> > To investigate this limitation, we replicated the MinMax Diffusion method on the ImageWoof dataset with IPC=10. Following the original approach, we fine-tuned the model, sampled 10 images per class, and evaluated performance in the IPC=10 setting. However, our replication resulted in significantly lower accuracy—approximately $35.8 \pm 2.0$, compared to the reported $39.2 \pm 1.3$. Upon further analysis, we discovered that sampling 100 images and randomly selecting 10 allowed us to match the reported results. This insight revealed that MinMax Diffusion does not consistently guarantee sample diversity during generation.
> >
> > To address this limitation, we developed a mode guidance method that explicitly conditions sample generation on distinct modes. This approach adds a level of control by ensuring diversity directly at the generation stage. Importantly, mode guidance is practical and effective, requiring no additional fine-tuning. Our method consistently generates diverse and representative samples, as qualitatively demonstrated in Figures 3, 5, and 6, and quantitatively validated in Table 9, where it surpasses previous methods in diversity metrics.
> >
> > Our motivation for mode guidance arose from the observed shortcomings of MinMax Diffusion and the need for a more robust sampling mechanism to guarantee diversity.
> >
> > # W5.2 - Clustering parameters
> > No additional parameters are used to compute the mode that can affect the performance. See above W.2.2 for more details.
> >
> > # W6 - Dataset Pruning Experiment
> >
> > We evaluated Dataset Pruning with Dynamic Uncertainty on ImageNette with IPC=10, 20, and 50. Table 1 evaluation was with hard labels; we also evaluated Dynamic Uncertainty with Hard Labels. Dynamic Uncertainty was computed by training a ResNet50 on ImageNette with 300 epochs. The dataset was built by choosing the top k highest uncertain samples per class, where k equals the IPC. Below is Table 1, including Dynamic Uncertainty results.
> >
> > | IPC                                  | 10                | 20                | 50                |
> > | ------------------------------------ | ----------------- | ----------------- | ----------------- |
> > | Random                               | 54.2$_{±1.6}$     | 63.5$_{±0.5}$     | 76.1$_{±1.1}$     |
> > | Dynamic Uncertainty                  | 54.2$_{±1.4}$     | 59.9$_{±0.5}$     | 69.1$_{±0.8}$     |
> > | DM                                   | 60.8$_{±0.6}$     | 66.5$_{±1.1}$     | 76.2$_{±0.4}$     |
> > | MinMaxDiff                           | 62.0$_{±0.2}$     | 66.8$_{±0.4}$     | 76.6$_{±0.2}$     |
> > | LDM                                  | 60.3$_{±3.6}$     | 62.0$_{±2.6}$     | 71.0$_{±1.4}$     |
> > | LDM+ Disentangled Diffusion (D$^4$M) | 59.1$_{±0.7}$     | 64.3$_{±0.5}$     | 70.2$_{±1.0}$     |
> > | **LDM+ MGD$^3$ (Ours)**              | **61.9$_{±4.1}$** | **65.3$_{±1.3}$** | **74.2$_{±0.9}$** |
> > | DiT                                  | 59.1$_{±0.7}$     | 64.8$_{±1.2}$     | 73.3$_{±0.9}$     |
> > | DiT+ Disentangled Diffusion (D$^4$M) | 60.4$_{±3.4}$     | 65.5$_{±1.2}$     | 73.8$_{±1.7}$     |
> > | **DiT+ MGD$^3$ (Ours)**              | **66.4$_{±2.4}$** | **71.2$_{±0.5}$** | **79.5$_{±1.3}$** |
> >
> >
> >
> >
> > # W7 - Concise version of our Contributions
> >
> > To summarize our contributions and results more concisely.
> > 1. We present a novel approach for dataset distillation by leveraging a pre-trained diffusion model to construct a synthetic dataset. (Contribution)
> >
> > 2. We introduce Mode Guidance with Stop Guidance to effectively control the denoising process, which enhances intra-class diversity and improves class representativeness, as demonstrated in Figures 3 and 5.  (Contribution)
> >
> > 3. Our method showcases improved or comparable performance in relation to the current state-of-the-art, without necessitating fine-tuning of the diffusion model. Notably, our approach achieves competitive outcomes even when employing a text-to-image model like Stable Diffusion, underscoring its adaptability and effectiveness across various diffusion models.  (Results)

---

> > > ### Comment · Reviewer_JiWQ · 2024-11-25
> > >
> > > Thank you for replying to my questions.
> > > It is recommended to revise the PDF accordingly, as many of my suggestions pertain to presentation and formatting. For example, the references currently start on a new page, which could be adjusted for better organization.
> > >
> > > Another question concerns the performance of Dynamic Uncertainty. The dataset pruning method appears to perform significantly worse than the random baseline in your results. However, according to their original paper, the performance of this method is reported to be significantly better than the random baseline. Could you clarify the reason for this discrepancy?
> > >
> > > Additionally, as I previously mentioned, what would the performance be if the dataset pruning method were combined with soft labels?

---

> > > > ### Author Response · Authors · 2024-11-26
> > > >
> > > > Thank you for your suggestions to improve the presentation and formatting of the manuscript. In response, we have revised the abstract, condensed the discussion of limitations in the introduction, streamlined the related works section, and clarified the caption for Figure 1. Additionally, we addressed the formatting issues, including ensuring that the references do not start on a separate page for better organization.
> > > >
> > > >
> > > > **Clarifying the discrepancy in Dynamic Uncertainty's performance compared to the original paper:**
> > > > Dynamic Uncertainty was evaluated on Dataset Pruning Setup, where the most extreme case evaluated on Dataset Pruning was about 50% of the dataset, which can be around IPC 650 in ImageNette. In our experiment, we evaluated Dynamic Uncertainty on a more extreme setup with IPC 10, 20, and 50. Surprisingly, Dynamic Uncertainty achieved a lower score for Dynamic than choosing N random images where N = IPC. However, Dynamic Uncertainty chooses the more uncertain samples in the original dataset, which is equivalent to hard samples. In the dataset distillation[1,2,3], literature has reported many times that building a distilled dataset using exclusively hard samples is not ideal; in fact, various previous works have recommended the opposite. Prioritizing easy samples in Low IPC settings (10, 100) and adding hard samples in higher IPC settings (e.g. 1000 and 2000). We believe that this is the case when evaluating Dynamic Uncertainty on Nette with Hard Labels.
> > > >
> > > >
> > > >
> > > > [1] Guo, Ziyao, et al. "Towards lossless dataset distillation via difficulty-aligned trajectory matching." ICLR (2024).
> > > >
> > > > [2]  Mirzasoleiman, B.,  et al. Coresets for Data-efficient Training of Machine Learning Models. ICML (2020)
> > > >
> > > > [3] Wang, Shaobo, et al. "Not all samples should be utilized equally: Towards understanding and improving dataset distillation." arXiv(2024).
> > > >
> > > >
> > > > **Dataset pruning method with soft labels:**
> > > >
> > > > We evaluated the Dynamic Uncertainty and our method using the Soft-label protocol on Nette. Below is the Table with our results.
> > > >
> > > > | Method  | IPC 10              | IPC 20              | IPC 50              |
> > > > |---------|---------------------|---------------------|---------------------|
> > > > | Ours    | 63.53 ± 2.04        | 74.73 ± 2.61        | 85.53 ± 0.46        |
> > > > | DynUnc  | 61.20 ± 0.72        | 72.33 ± 2.05        | 85.07 ± 0.46        |

---

> > > > > ### Author Response · Authors · 2024-11-29
> > > > >
> > > > > We deeply appreciate the time and thoughtful effort you have invested in reviewing our manuscript. We have carefully addressed your comments in our rebuttal and are committed to resolving any remaining concerns. We kindly request your timely feedback to ensure we can provide comprehensive clarifications before the deadline.

---

> > > > ### Author Response · Authors · 2024-12-02
> > > >
> > > > Thank you for all the suggestions and feedback. We have diligently taken up the suggestions and carefully updated the pdf. Also, we answered the queries in as much detail as possible. As the deadline for author discussion is approaching close, we would be happy to clarify if there are any further questions or concerns. Thank you !

---

### Meta-Review · Area_Chair_zkDy · 2024-12-21

**Metareview:**

This paper presents a pipeline for data set distillation with a pretrained diffusion model: Mode Discovery, Mode Guidance, and Stop Guidance. It shows promising results compared to baselines dataset distillation methods. The paper receives mixed ratings: while reviewers acknowledge that the method makes sense and delivers promising results, concerns are raised regarding clarity and results. I believe that the authors did a great job responding to the concerns, yet regretfully not all reviewers are fully convinced. To me this is a really a borderline paper. It appears that the writing does need to be improved regarding its motivation and positioning against prior works, it can benefit from clearly ablating some key components such as the role of the clustering algorithm. Based on these considerations, I'll recommend reject for this one but encourage the authors to submit it to the next cycle.

**Additional Comments On Reviewer Discussion:**

Concerns are raised regarding clarity and results. I believe that the authors did a great job responding to the concerns, yet regretfully not all reviewers are fully convinced

---

### Decision · Program_Chairs · 2025-01-22

Reject